# τ-BENCH: A BENCHMARK FOR TOOL-AGENT-USER INTERACTION IN REAL-WORLD DOMAINS

**Shunyu Yao**[*]    **Noah Shinn**    **Pedram Razavi**    **Karthik Narasimhan**

Sierra

## ABSTRACT

Existing benchmarks do not test language agents on their interaction with human users or ability to follow domain-specific rules, both of which are vital for deploying them in real world applications. We propose τ-bench, a benchmark emulating dynamic conversations between a user (simulated by language models) and a language agent provided with domain-specific API tools and policy guidelines. We employ an efficient and faithful evaluation process that compares the database state at the end of a conversation with the annotated goal state. We also propose a new metric (pass^k) to evaluate the reliability of agent behavior over multiple trials. Our experiments show that even state-of-the-art function calling agents (like `gpt-4o`) succeed on $< 50\%$ of the tasks, and are quite inconsistent (pass^8 $< 25\%$ in retail). Our findings point to the need for methods that can improve the ability of agents to act consistently and follow rules reliably.

## 1 INTRODUCTION

There is increasing excitement around the potential of language agents (Sumers et al., 2023; Yao et al., 2022; Schick et al., 2023; Ahn et al., 2022) to enable new levels of automation across various industries. However, their deployment in real-world systems requires several key desiderata to be satisfied. Agents must (1) interact seamlessly with both humans and programmatic APIs over long horizons to incrementally gather information and resolve intents, (2) accurately adhere to complex policies and rules specific to a task or domain, and (3) maintain consistency and reliability at scale, across millions of interactions. For instance, consider the case of an airline booking agent (Figure 1). When a user wants to change their flight reservation to a different destination airport, the agent needs to gather the required information by interacting with the user, check the airline policies using the guidelines provided, and find new flights and (if possible) rebook the user using complex reservation APIs. In addition, the agent should be consistent in its behavior across different kinds of users with the same request, and robust to small changes in the conversation flow that should not affect the end outcome.

Modeling realistic human interaction and rule following in agent evaluation is vital for developing and deploying trustworthy agents in the wild, and tackling challenges like long-context reasoning and planning in a methodical fashion. Existing benchmarks (Yao et al., preprint; Zhou et al., 2023; Jimenez et al., 2023; Liu et al., 2023; Ruan et al., 2023) for language agents often feature simplified instruction-following setups, where the agent autonomously interacts with an environment (web, code terminal, or APIs) given all the information upfront, without any human-in-the-loop interaction and without the need to consult any domain-specific guidelines.

In this work, we introduce τ-bench (short for Tool-Agent-User Interaction Benchmark) to measure an agent's ability to interact with (simulated) human users and programmatic APIs while following domain-specific policies in a consistent manner. τ-bench is built in a modular framework with (1) realistic databases and APIs, (2) domain-specific policy documents, and (3) instructions for diverse user scenarios and corresponding ground truth annotations. As a first demonstration, we focus on the realm of customer service and create two different domains where agents need to assist

---

[*]Work done during internship. Code and data: `https://github.com/sierra-research/tau-bench`.

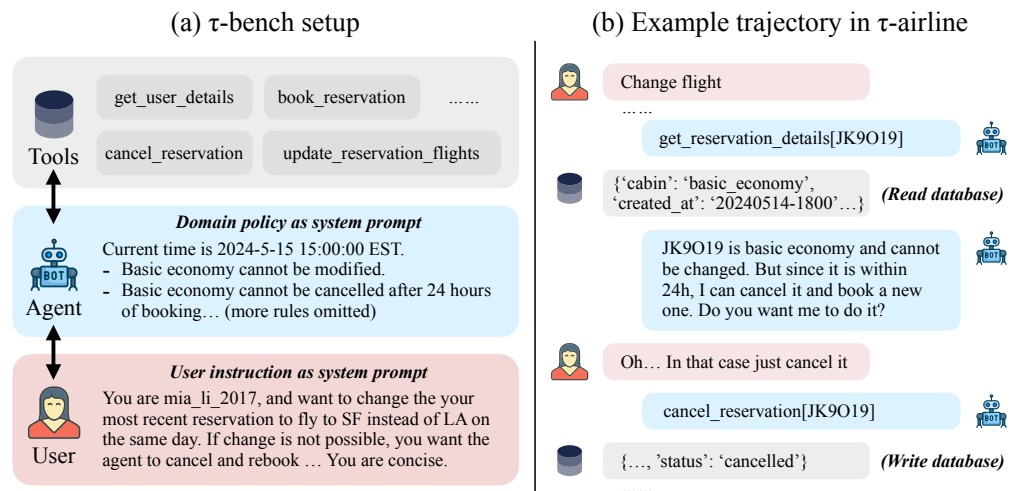

(a) τ-bench setup  (b) Example trajectory in τ-airline

Figure 1: (a) In τ-bench, an agent interacts with database API tools and an **LM-simulated user** to complete tasks. The benchmark tests an agent's ability to collate and convey all required information from/to users through multiple interactions, and solve complex issues on the fly while ensuring it **follows guidelines** laid out in a domain-specific policy document. (b) An example trajectory in τ-airline, where an agent needs to reject the user request (change a basic economy flight) following domain policies and propose a new solution (cancel and rebook). This challenges the agent in long-context zero-shot reasoning over complex databases, rules, and user intents.

simulated users with diverse requests (τ-*retail* and τ-*airline*). We leverage the generative capabilities of language models (LMs) for data creation and realistic human user simulation (Park et al., 2023) in conjunction with manual annotation and verification.

We constructed τ-bench in three stages, including manual schema and API design, LM-assisted generation of data entries, and manual scenario generation and verification for the user simulator. Our evaluation scheme compares the database state at the end of each episode with the ground truth expected state. This allows for objective measurement of the agent's decision making, while also providing room for stochastic variation in the conversation itself, since the user may pose the same request in different ways that result in the same end state of the database. We also introduce the metric of pass^k, which measures the consistency and robustness of the agent across $k$ i.i.d. trials.

Our experiments reveal that agents built with simple LM constructs (like function calling or ReAct) perform poorly, highlighting the need for more sophisticated agent architectures. For instance, even state-of-the-art LMs like `gpt-4o` achieve low task success rates (pass^1) using function calling (∼61% on τ-retail and ∼35% on τ-airline). With increasing $k$, the chance of consistently solving a task drops rapidly, to as low as ∼25% for pass^8 on τ-retail for the same model. This showcases the fragile nature of such agents in handling stochasticity and partial information, which is common in human-agent interaction. Upon analyzing the failure cases, we find that current agents struggle with complex reasoning over databases, understanding and following ad-hoc policies, and handling compound (more than one) requests. We hope that τ-bench enables the evaluation and development of more consistent and capable agents for real-world digital tasks involving human interaction.

## 2  RELATED WORK

Most existing benchmarks for agents and task-oriented dialogue systems focus on evaluating either conversational or tool-use capabilities. τ-bench aims to unify both under realistic settings, while also testing how well agents can follow domain-specific policies in a consistent manner.

**Benchmarks for language agents and tool use.** Several benchmarks have been developed to evaluate agents powered by LMs (Yao et al., preprint; Zhou et al., 2023; Jimenez et al., 2023; Liu et al., 2023; Ruan et al., 2023) . Recent efforts have focused specifically on evaluating tool use capabilities of LMs, i.e., their ability to generate the right function calls from a set of functions in an API. Projects like the Berkeley Function Calling Leaderboard (BFCL)  (Yan et al., 2024), Tool-

Bench (Xu et al., 2023) and MetaTool (Huang et al., 2023) test tool use/function calling in multiple programming languages and propose various methods for evaluating the accuracy of function calls. ToolEmu (Ruan et al., 2023) uses language models themselves to emulate the execution of tools, with a focus on exposing potential safety risks when LM agents fail to use tools correctly. However, all these works only contain a single-step user interaction, where the human interacting with the agent provides an initial instruction containing all the required information. In contrast, our benchmark focuses on a more realistic setting where the agent has to interact with human users to gather information and authorization.

**Task-oriented dialogue.** Task-oriented dialogue has been a long-standing challenge for NLP, with several efforts over the years to build domain-specific offline datasets or user simulators. The former types of benchmarks are static and only test the conversational agent on pre-collected conversation trajectories(Chen et al., 2021a; Budzianowski et al., 2018; Andreas et al., 2020). The latter either rely on user simulators that are rule-based or rely on symbolic specifications (Schatzmann et al., 2007; Gür et al., 2018) or perform tests with real humans through crowdsourcing platforms (He et al., 2018). Some very recent work explores the use of LMs as response raters to train dialogue systems (Hu et al., 2023) or evaluates their capability for simulating users (eun Yoon et al., 2024). $\tau$-bench leverages the powerful text generation capabilities of state-of-the-art LMs to simulate realistic user utterances and long-context conversations using *textual* scenario descriptions, with the goal of evaluating agents. The stochastic sampling from LMs allows for diverse yet faithful variations in the dialogue when re-run with the exact scenario – this is extremely useful for testing agent consistency, as we show in § 5.1.

**User simulation with LMs.** Our work is also related to recent efforts on using LMs as simulators of human characters. This includes papers on simulating non-player characters (NPCs) in text adventure games (Kim et al., 2022) multiple agents in human-like societies (Park et al., 2023) or specific collaborative tasks (Wu et al., 2023), and enabling human-in-the-loop interaction for tasks like online shopping (Chen et al., 2024) or web search (Zhang et al., 2024). However, all these past works have not used such simulators to benchmark the reliability of agents, instead focusing on showcasing the ability of LMs to enable realistic simulations. Our work uses such realistic user simulation to provide an accurate assessment of the reliability and robustness of AI agents for deployment in systems that undertake millions of real-world interactions with humans.

# 3 $\tau$-BENCH: A BENCHMARK FOR TOOL-AGENT-USER INTERACTION

Each individual task in $\tau$-bench can be formulated as a partially observable Markov decision process (POMDP) $(\mathcal{S}, \mathcal{A}, \mathcal{O}, \mathcal{T}, \mathcal{R}, \mathcal{U})$ with state space $\mathcal{S}$, action space $\mathcal{A}$, observation space $\mathcal{O}$, transition function $\mathcal{T} : \mathcal{S} \times \mathcal{A} \rightarrow \mathcal{S} \times \mathcal{O}$, reward function $\mathcal{R} : \mathcal{S} \rightarrow [0, 1]$, and instruction space $\mathcal{U}$. The agent interacts with both (1) databases ($db$) via API tools, and (2) a (simulated) user ($user$) to complete a task, i.e., $\mathcal{S} = \mathcal{S}_{db} \otimes \mathcal{S}_{user}$, $\mathcal{A} = \mathcal{A}_{db} \cup \mathcal{A}_{user}$, $\mathcal{O} = \mathcal{O}_{db} \cup \mathcal{O}_{user}$. In addition, the agent is provided a domain-specific policy document containing rules it must adhere to – one can think of this as partially describing the world model of the domain. We describe each component in more detail below.

**Databases and APIs.** Each $\tau$-bench domain has several databases and associated APIs. The contents of the database form the state $s_{db}$ (Figure 2a), which is hidden from the agent and the user, and can only be read from or written to using API actions $a_{db}$, which are usually in the form `tool_name(**kwargs)`. When an action is executed on the database, the transition $\mathcal{T}_{db} : (s_{db}, a_{db}) \mapsto (s'_{db}, o_{db})$ is deterministic and implemented as a Python function (Figure 2b).

**Domain policy.** Each domain has a policy (Figure 2c) that explains the domain databases, task procedures, and restrictions for the agent to follow in its interactions. Some restrictions are implemented as checks in the API, e.g., using a payment ID not in the user profile will lead to $o_{db} =$ `"Error: payment not found"`, and others not, e.g., the airline policy states different baggage allowances for different membership statuses and cabin classes, but the agent needs to fill in the number of baggage items to be paid for in the `book_reservation` API, similar to the freedom given real-world agents.

**User simulation.** We use a language model (`gpt-4-0613`) to simulate a human user interacting with the agent. The user state $s_{user}$ consists of an initial system prompt with the task instruction

```json
{"order_id": "#W2890441",
"user_id": "mei_davis_8935",
"items": [{
    "name": "Water Bottle",
    "product_id":
    "8310926033",
    "item_id": "2366567022",
    "price": 54.04,
    "options": {
        "capacity": "1000ml",
        "material": "stainless
        steel",
        "color": "blue"
    }}, ...], ...}
```

(a) An `orders` database entry in $\tau$-retail.

```
## Return delivered order
- After user confirmation, the order
status will be changed to 'return
requested'...

## Exchange delivered order
- An order can only be exchanged if its
status is 'delivered'...
```

(c) Domain policy excerpts in $\tau$-retail.

```python
def return_delivered_order_items(
    order_id: str,
    item_ids: List[str],
    payment_method_id: str,
) -> str: ...

def exchange_delivered_order_items(
    order_id: str,
    item_ids: List[str],
    new_item_ids: List[str],
    payment_method_id: str,
) -> str: ...
```

(b) An API tool in $\tau$-retail.

```json
{"instruction": "You are Mei Davis in
80217. You want to return the water
bottle, and exchange the pet bed and
office chair to the cheapest version.
Mention the two things together. If you
can only do one of the two things, you
prefer to do whatever saves you most
money, but you want to know the money
you can save in both ways. You are in
debt and sad today, but very brief.",
"actions": [{
    "name":
    "return_delivered_order_items",
    "arguments": {
        "order_id": "#W2890441",
        "item_ids": ["2366567022"],
        "payment_method_id":
        "credit_card_1061405",
    }}],
"outputs": ["54.04", "41.64"]}
```

(d) User instruction ensures only one possible outcome.

Figure 2: $\tau$-bench is constructed in a modular fashion with several components: (a) JSON databases, (b) Python API tools, (c) Markdown domain policies, and (d) JSON task instances. The agent can only access API tools and domain policies, and indirectly access databases via API tools. Task annotation is **not visible to the agent** and is used only for user simulation and evaluation.

(Figure 2d) along with the entire conversation history between the user and the agent so far. The user cannot see the interaction history between the agent and API tools. The agent can interact with the user using any natural language message, e.g., $a_{user}$ can be `"Your reservation has been updated, is there anything else I can help with?"`. The transition $\mathcal{T}_{user}$ : $(s_{user}, a_{user}) \mapsto (s'_{user}, o_{user})$ is stochastic and attaches the agent's message to the chat history followed by sampling a new user message from the LM, e.g., $o_{user}$ can then be `"Yes, I also want to cancel another flight."` When the user issues $o_{user}$=`"###STOP###"`, the episode finishes and the agent is evaluated.

**Task instances.** As shown in Figure 2d, each $\tau$-bench task instance has two parts: an instruction for the user simulation (hidden from agents), and an annotation of the ground truth database write actions (and optionally, ground truth outputs for user questions). The instruction sets up user identity, intent, and preferences in a way that guarantees only one possible outcome under the domain policy. Each task episode consists of the simulated user starting with a request, which the agent handles in a conversational manner while being able to call tools at any point and refer to the provided policy. Once the episode ends, the database state and agent-to-user messages are used to compute the reward.

**Reward.** The reward of a task episode $r = r_{\text{action}} \times r_{\text{output}} \in \{0, 1\}$ is based on (1) whether the final database is identical to the unique ground truth outcome database ($r_{\text{action}}$), and (2) whether the agent's responses to the user contain all necessary information ($r_{\text{output}}$). So for the task of Figure 2d, the agent-user dialogue can be varied and the agent can call various (read) actions, but the agent is successful if the only database write action is `return_delivered_order_items(order_id="#W2890441",`

`item_ids=["2366567022"], payment_method_id="credit_card_1061405")`, and the user responses contain `"54.04"`, `"41.64"` as substrings. Note that $r = 1$ might be a necessary but not sufficient condition for a successful episode e.g., the agent might issue the return without explicit user confirmation, which violates the policy. Nevertheless, our proposed rule-based reward is fast to compute and faithful, and already poses significant challenges for current models and methods as we show in § 5.

**Passˆk metric.** For tasks like code generation with good verification techniques (unit tests), the community has defined the pass@k (pass at k) metric as the chance that at least one out of $k$ i.i.d. task trials is successful, which captures the trend of agents enabling *discovery* of solutions with scaling of inference-time compute (Chen et al., 2021b). For real-world agent tasks requiring *reliability and consistency* like customer service, we propose a new metric – passˆk (pass hat k), defined as the chance that *all* $k$ i.i.d. task trials are successful, averaged across tasks. Therefore, if a task is run for $n$ trials and $c$ of those trials end up successful ($r = 1$), unbiased estimates for passˆk and pass@k would be:

$$\text{pass\^{}k} = \mathbb{E}_{\text{task}} \left[ \binom{c}{k} \Big/ \binom{n}{k} \right], \quad \text{pass@k} = 1 - \mathbb{E}_{\text{task}} \left[ \binom{n-c}{k} \Big/ \binom{n}{k} \right].$$

In our case, for the same task, the user prompt and database transitions are the same, with just the LM sampling of the user and agent messages generating sufficient stochasticity. Thus, passˆk can capture the reliability of the agent at handling variations in conversations with the same underlying semantics while adhering to the domain policies and rules. By default, we report the average reward across tasks, passˆ1=pass@1=$\mathbb{E}[r] = \mathbb{E}[c/n]$, as the main metric for comparing agents.

## 4 BENCHMARK CONSTRUCTION

$\tau$-bench defines domain-agnostic environment and user simulation classes shared by various domains, and domain-specific data in terms of database JSON files, database API Python code and documentation, domain policy text, and task instances. Each domain is created in a three-stage approach with a mix of LM and code runs, and human labeling and checking.

**Stage I: Manual design of database schema, APIs, and policies.** We start by co-designing the simplest possible database schemas, APIs, and policies with inspiration (and simplification) from their real-world counterparts. Simplicity is important for the logical consistency of various components and the ease of API and task annotation. Still, a minimally realistic domain requires at least tens of schemas, APIs, rules, and turns out to be challenging enough for existing agents. See § B.1 for more.

**Stage II: Automatic data generation with LMs.** Once the data schema is set up, we create an example entry and use gpt-4 to generate a systematic code snippet to sample scalable entries, and manually polish minor bugs in the code. See § B.2 for an example snippet and more details.

**Stage III: Manual task annotation and validation with agent runs.** Here, the key challenge is to ensure the user instruction leads to a unique database outcome. For example, if the preferred payment method is not specified, the user might answer differently and cause the final database to be different across trials. So we write an initial user instruction, run a trial with `gpt-4-turbo` function calling agent, polish the user instruction by examining the trajectory, and do this iteratively until we are certain no ambiguities exist (see Figure 7 in § A, where we run each $\tau$-retail task with $> 40$ `gpt-4-turbo` trials and check all tasks with zero or low success rates). We can copy and edit agent actions and outputs for ground truth annotation, which is easier than annotating from scratch.

In practice, we might update minor details of database schemas or policies during data or task creation, but the three stages are mostly linear, and the constructed data is organized in a modular structure.

### 4.1 DOMAINS

Using the above procedures, we modularly construct two domains, $\tau$-retail and $\tau$-airline. We choose these two domains as they are relatively easy to synthesize data (e.g., products, prices, flights) and

|  | $\tau$-retail | $\tau$-airline |
|---|---|---|
| **Databases** | 500 users, 50 products, 1,000 orders | 500 users, 300 flights, 2,000 reservations |
| **API tools** | 7 write, 8 non-write | 6 write, 7 non-write |
| **Tasks** | 115 | 50 |

Table 1: Key statistics from $\tau$-retail and $\tau$-airline.

craft policies (e.g., product return, baggage allowance) based on common sense, allow for diverse tasks, and are close to real-world applications. For more capable agents in the future, more advanced domains (e.g., medical, tax, or legal) with more complex data and rules can be studied. Below, we briefly describe the domain policies of two domains (full details of the domains in § B.1).

$\tau$**-retail.** In this domain, the agent is tasked with helping users cancel or modify pending orders, return or exchange delivered orders, modify user addresses, or provide information. Each product (e.g., "Water Bottle" in Figure 2a) has various item options with unique IDs (e.g., 1000ml, stainless steel, blue). Each pending order can only be canceled or modified once, and each delivered order can only be returned or exchanged once. An item cannot be modified or exchanged for another product type. These constraints simplify task and API design, and challenge agents to follow domain-specific rules, and inform and collect complete information from users before taking actions.

$\tau$**-airline.** Here, the agent has to help users book, modify, or cancel flight reservations, or provide refunds. We construct 300 flights between 20 US cities with realistic durations and prices, and API tools to query direct or one-stop flights. The domain policy is more complex than $\tau$-retail, with ad-hoc constraints about combining payment methods, checked bag allowance, flight changes and cancellations, etc. These constraints can also be over membership tier and cabin class specific, creating challenging multi-hop reasoning puzzles for the agent.

## 4.2 KEY CHARACTERISTICS

**Realistic dialogue and tool use.** Compared to prior task-oriented dialogue benchmarks, $\tau$-bench has more complex databases and realistic user simulations thanks to the advances of LMs. Some trajectories can be seen in § C.2 and § D.2. Notably, even if the user instruction is synthetic, the user utterances generated via LMs are open-ended and natural-sounding.

**Open-ended and diverse tasks.** Each $\tau$-bench domain's data schemas, APIs, and rules are simplified compared to real-world domains, but they are rich enough to support the creation of extremely diverse, open-ended, and sometimes creative tasks (see § A, § C.2, § D.2). Importantly, we trade off quantity for quality — as § 5 shows, running a small set of high-quality tasks for multiple trials (with pass^k metric) can reliably reveal rich insights into different models, methods, and research challenges.

**Faithful rule-based evaluation.** Real-world agents are hard to evaluate as the trajectory can be extremely diverse for the same task, and success criteria are multi-faceted. As a result, it often requires human evaluation, e.g., end users to judge task resolution and domain experts to judge rule following. In $\tau$-bench, we trade off slow, careful task annotation for fast, faithful evaluation. By ensuring that only one database outcome is possible based on domain policies and user desires, subjective and noisy human judgments can be replaced by simple and objective database state comparisons.

**Modular extension.** The codebase structure of $\tau$-bench is modular, and it is easy to add new domains to $\tau$-bench, or add or update database entries, domain functionalities, rules, APIs, tasks, and evaluation metrics (given they are consistent with the existing domain data). We release our codebase publicly to encourage the community to create new tasks and domains for $\tau$-bench.

## 5 EXPERIMENTS

**Models.** We test various state-of-the-art proprietary and open language models for agents through their APIs: OpenAI GPT API (`gpt-4o`, `gpt-4-turbo`, `gpt-4-32k`, `gpt-3.5-turbo`), Anthropic Claude API (`claude-3-opus`, `claude-3-sonnet`, `claude-3-haiku`), Google Gemini API (`gemini-1.5-pro-latest`, `gemini-1.5-flash-latest`),

| Model | retail | airline | avg |
|---|---|---|---|
| `gpt-4o` | **61.2** | **35.2** | **48.2** |
| `gpt-4-turbo` | 57.7 | 32.4 | 45.1 |
| `gpt-4-32k` | 56.5 | 33.0 | 44.8 |
| `gpt-3.5-turbo` | 20.0 | 10.8 | 15.4 |
| `claude-3-opus` | 44.2 | 34.7 | 39.5 |
| `claude-3-sonnet` | 26.3 | 27.6 | 27.0 |
| `claude-3-haiku` | 19.0 | 14.4 | 16.7 |
| `gemini-1.5-pro` | 21.7 | 14.0 | 17.9 |
| `gemini-1.5-flash` | 17.4 | 26.0 | 21.7 |
| `mistral-large` | 30.7 | 22.4 | 26.6 |
| `mixtral-8x22b` | 17.7 | 31.6 | 24.7 |
| `meta-llama-3-70B` | 14.8 | 14.4 | 14.6 |

Table 2: Pass^1 across models via function calling, except Llama-3 via text-ReAct. Average is weighted by domains, not by tasks.

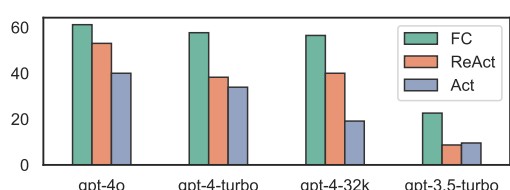

Figure 3: pass^1 across models/methods in $\tau$-retail.

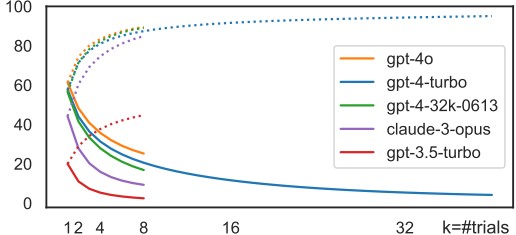

Figure 4: pass^k (–) and pass@k (..) in $\tau$-retail.

Mistral API (`mistral-large`, `open-mixtral-8x22b`), AnyScale API (`meta-llama-3-70B-instruct`). Only the last two models openly release weights. We do not test small models (7/13B) due to the difficulty of the benchmark.

**Methods.** Our main method for building the agent is through the use of function calling (FC), which is natively supported by all tested LMs except Llama-3. In FC mode, the model's system prompt is set to be the domain policy, and at each turn, the model autonomously decides to generate a user response message or a tool call. We also test text-formatted ReAct (Yao et al., 2023b) and its Act-only ablation, where the model is instructed to zero-shot generate "Thought: {some reasoning} Action: {some JSON format action argument}" or only the action part. Notably, some agent methods are not suitable for a user-in-the-loop setup, e.g., self-reflection (Shinn et al., 2023) is unrealistic as real-world agents only have one chance to serve the user, and planning approaches (Yao et al., 2023a) might be too slow to help a user in real time.

We limit each task to at most 30 agent actions (either tool calls or user responses). For main results (Table 2), we run at least 3 trials per task. The LM temperature is 0.0 for agent and 1.0 for user.

## 5.1 MAIN RESULTS

**Model comparison.** From Table 2, we see that `gpt-4o` is the best model with function calling, and there is a wide spectrum of performances among various models. Notably, SoTA open-weight models (llama-3-70b and mistral-8x22b) still have a significant gap to cover with respect to SoTA proprietary models (gpt-4o, claude-3-opus). All models are still far from solving $\tau$-bench, especially the more challenging $\tau$-airline where even gpt-4o solves only $35.2\%$ of the tasks. The diversity of model performances (shown in Table 2) and task difficulties (shown in Figure 7 in § A) as well as large remaining gaps from perfect resolution makes $\tau$-bench ideal for benchmarking and developing new models for agents, tool use, and dialogue.

**Method comparison.** Figure 3 shows that natively supported function calling consistently outperforms text-formatted agent methods with the state-of-the-art models. For text-formatted agent methods, adding reasoning traces still consistently helps (compare ReAct vs. Act columns) as it helps bridge the gap between observations and actions that have unfamiliar formats. We have also experimented with adding a "think" function for function-calling agents, but it did not boost performance, perhaps because most FC models have not been trained toward such reasoning.

**Agent consistency via pass^k.** As shown in Figure 4, the chance of reliably and consistently solving the same task multiple times significantly drops as the number of trials $k$ increases. Even for the best-performing gpt-4o function calling agent which has a $> 60\%$ average task success, pass^8 drops

to $< 25\%$. In real-world scenarios, it is important and challenging not just to build agents with high average success (pass^1), but with more robustness and consistency (pass^k trend).

**Cost analysis.** When we pair `gpt-4o` FC agent with `gpt-4` user simulation on $\tau$-retail, the agent / user simulation costs are \$0.38 / \$0.23 per task respectively, so running one trial per task costs around 200 dollars. For the agent, the input prompt / completion output take up $95.9\%$ / $4.1\%$ of the price respectively, so the cost is mainly due to long system prompt (domain policy + function definitions).

## 5.2 RESEARCH CHALLENGE ANALYSIS

In this subsection, we analyze in both quantitative and qualitative terms the challenges of $\tau$-bench, with a focus on the $\tau$-retail split and the most advanced baseline: `gpt-4o` function calling agent.

**Failure breakdown.** We sample 115 gpt-4o FC agent trajectories in $\tau$-retail (1 trial per task), out of which 40 tasks have failed (pass^1=65.2%). Upon manual examination of these failures, 4 of them are caused by user instruction typo or ambiguity (and then fixed), and the remaining 36 failure cases are agent issues, which are broken down into more detail below and in Figure 5.

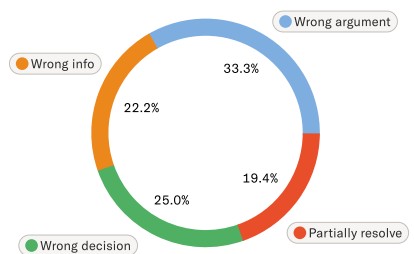

Figure 5: Breakdown of 36 failed `gpt-4o` FC agent trajectories in $\tau$-retail.

**Failure 1: Wrong argument or information provided: the challenge of complex database reasoning.** For "wrong argument", `gpt-4o` FC agent usually makes the right type of tool call(s) but fills in one or more arguments incorrectly. In the example shown in § C.2.2, the user wants to exchange a lamp for a less bright one and prefers an AC adapter over battery or USB power source. The agent fails to reason over the complex inventory of lamps and find the unique option given such a preference. Weaker models and methods struggle with even more basic failures such as hallucinating arguments — for example, while `gpt-4o` FC agent only makes $0.46$ tool calls with non-existent user/product/order/item IDs per $\tau$-retail task, `gpt-3.5-turbo` FC / Act agents make $2.08$ / $6.34$, respectively.

For "wrong info", agents omit user-required information (e.g., the user asks for a tracking ID but the agent does not provide it), or calculate the wrong information (e.g., wrong total price), or provide the user with incorrect information that causes the user request to diverge (e.g., the user might cancel or exchange based on incorrect price information provided by the agent). These failures account for ~55% of overall failures and highlight the need for improved common sense and numerical reasoning over complex databases and user intents for future models.

**Failure 2: Incorrect decision-making: the challenge of domain understanding and rule following.** While the above failures can be recognized even without referring to the domain policy, "wrong decision-making" failures (25% of overall failures) occur as the agent fails to understand the domain-specific knowledge or rules and makes the wrong type of tool call. In the example of § C.2.1, the user wants to exchange "a couple of items", and according to the domain policy, "Exchange or modify order tools can only be called once. Be sure that all items to be exchanged are collected into a list before making the tool call". However, the gpt-4o FC agent omits the domain knowledge and rule and decides to exchange one item first, resulting in the second item not being exchanged.

To further understand how different agents follow rules in different domains, we perform an ablation study by removing the domain policy from the FC agent system prompt. As seen in Table 3, in $\tau$-retail where rules are simpler and closer to commonsense, `gpt-4o` and `gpt-3.5-turbo` agents only degrade $4.4\%$ and $5.5\%$ in terms of pass^1, suggesting that their successful cases mostly stem from using tools in an intuitive and common sense way, and that they may not actually be leveraging

|  | $\tau$-retail | $\tau$-airline |
|---|---|---|
| `gpt-4o` | $61.2 \rightarrow 56.8$ | $\mathbf{33.2 \rightarrow 10.8}$ |
| `gpt-3.5` | $20.0 \rightarrow 14.5$ | $10.8 \rightarrow 9.6$ |

Table 3: pass^1 scores degrade when the domain policy is not provided in the agent's system prompt.

| User agent strategy | Acc | Errors caused by the agent | Errors caused by the user | Other |
|---|---|---|---|---|
| llm | 0.367 | 0.980 | 0.020 | 0.000 |
| react | 0.300 | 0.880 | 0.020 | 0.100 |
| verify | 0.393 | 0.960 | 0.040 | 0.000 |
| reflection | 0.406 | 0.920 | 0.040 | 0.040 |

Table 4: Analysis performed on trajectories from 3 trials on the Airline domain with gpt-4o function-calling as the model and agent strategy. Acc is reported as the average accuracy over 3 trials. Agent, user, and other errors are reported as proportions of 50 randomly sampled error trajectories.

the policy documents to the extent possible. In $\tau$-airline where rules are more complex and ad-hoc (e.g., baggage allowance varies for different membership tiers and cabins), removing the policy hurts `gpt-4o` significantly ($-22.4\%$) but `gpt-3.5-turbo` only slightly ($-1.2\%$), suggesting the former follows rules at times but the latter does not has the capacity to process complex airline rules. Overall, $\tau$-bench poses significant challenges for function calling agents to follow complex domain dynamics and rules, and showcases there is still work to be done in this direction. Domain-specific fine-tuning or agent code scaffolding might provide some remedy, which can be important future work.

**Failure 3: Partial resolution of compound requests.** Lastly, as shown in Figure 6, when a task involves many user requests (represented by the number of ground truth write actions to databases), it becomes more challenging for function calling agents (19% of cases). Sometimes the agent omits explicit user requests at the beginning of the conservation, hinting at the need for better long-context and memory capabilities. Other times, the agent omits implicit actions, such as in § C.2.3, where the user wants to fix wrong addresses in all orders, but the agent stops after checking only one order. Agents need to improve in their consistency and systematicity in handling such cases.

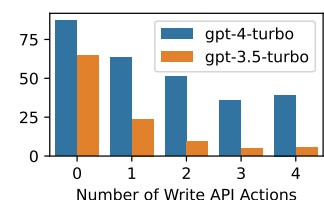

Figure 6: Retail tasks with more database writes are harder.

### 5.3 USER SIMULATION METHODS

One challenge of simulating users with a language model is that it could cause occasional hallucinations/misdirections that may cause episodes to fail even if the agent is not at fault. We explored alternative user simulation strategies to exhibit better instruction-following ability, drawing on recent advancements in agent development. We compared the vanilla strategy described above (llm) to three other strategies: **(1) react**: write a reasoning trace before responding (Yao et al., 2022), **(2) verify**: generate response proposals until an LLM verifier returns true, and **(3) reflection**: iteratively refine a response until the LLM verifier returns true (Shinn et al., 2023; Madaan et al., 2023). To measure the impact of hallucinations and other mistakes caused by each user strategy, we manually evaluate 50 randomly sampled *failed* trajectories from 3 trials over the entire Airline domain.

From Table 4, we find that all user simulation strategies perform relatively similarly, with reflection performing the best. Further, we find that less than 4% of errors are caused by the user simulator across all user strategies.

## 6 DISCUSSION

We have presented $\tau$-bench, a novel benchmark for evaluating the reliability of agents in interacting with humans and tools in dynamic and realistic settings. The benchmark leverages the latest advances in LMs to simulate users, allows for automated testing of agents and provides an assessment of an agent's ability to follow domain-specific rules in a consistent manner. Our results show that even SOTA LMs are far from being reliable for use in real-world settings.

**Directions for improvement.** While $\tau$-bench is a step towards dynamic evaluation of agents in real-world scenarios, there are several directions for improvement. The simulated user can have some limitations: (1) the user instruction might contain typos or ambiguities, which annotators can

examine and fix; (2) the user instruction may not contain all domain knowledge, e.g., in § C.2.1, the user authorizes the single item exchange without knowing that the agent could only issue one exchange action, which reflects real-world users who (rightfully) do not know complex domain policies; or (3) the user simulation LM might have limited capacity at reasoning, calculation, long-context memorization, or alignment with the instruction prompt, e.g., in § C.2.2 the user authorizes the agent-recommended lamp without double checking its features. While these can all be improved in future work, one can also argue that this is indicative of the real world where users can have a wide range of skill sets and knowledge, and the onus is on the agents to handle diverse users.

In addition, one can also add more systematic checks to the simulator to ensure unique outcomes. The domain policies can also be made more complex to match real-world scenarios. More evaluation metrics can be added to define agent success (e.g., LM checks that certain rules are followed). The manual annotation process for the benchmark is difficult and requires a deep understanding of both the domain and agent capabilities. There is also some element of implicit bias during the task curation process since we use the `gpt-4-turbo` FC agent to tune the user's system prompt. Future work can investigate alternative ways of using LMs for improving data curation and user simulation. Finally, while we don't believe this work has potential negative societal implications directly, it helps real-world agents which can have various consequences for the economy and society in the future.

**Challenges for agents.** At the core, the main results from our experiments demonstrate a critical fact: agents built on top of LM function calling lack sufficient consistency and rule-following ability to reliably build real-world applications. Solving both of these problems can have outsized impact on automating several real-world tasks and ensuring smoother human-in-the-loop interaction. Other specific features to improve in agents include long-horizon information tracking and memory, as well as the ability to focus on the right pieces of information in context for the decision at hand, especially when there may be conflicting facts present.

ACKNOWLEDGEMENTS

We thank Clay Bavor, Honghua Dong, Victor Barres, Dongxu Zhou, and Yangjun Ruan for feedback on earlier drafts of the paper, and Nate White for helping set up the different LLM APIs for the experiments.

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

## A    ADDITIONAL RESULTS

As shown in Figure 7, $\tau$-bench tasks have a well-balanced and diverse spectrum of difficulties. We use such a plot to find tasks with zero success, and examine the task annotations in a targeted way.

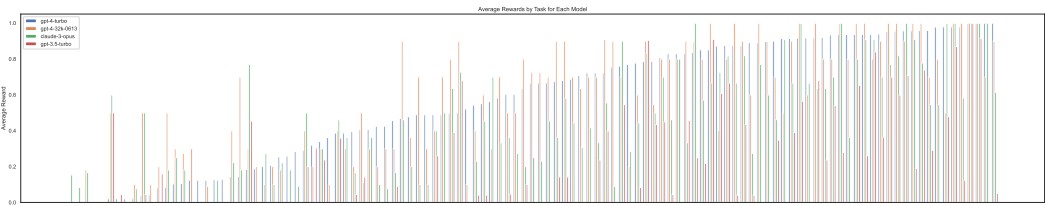

Figure 7: The success rate of each $\tau$-retail task, sorted by `gpt-4-turbo` success rate. Each task has at least 40 `gpt-4-turbo` trials to ensure reliable per-task success rates.

## B    BENCHMARK CONSTRUCTION

### B.1    STAGE I: DESIGN OF DATABASE SCHEMAS, APIS, AND POLICIES

The data schema examples can be seen in § C.1 and § D.1.

| | $\tau$-retail | $\tau$-airline |
|---|---|---|
| **Databases** | users, products, orders | users, flights, reservations |
| **Read APIs** | find_user_id_by_email
find_user_id_by_name_zip
list_all_product_types
get_order_details
get_product_details
get_user_details | get_reservation_details
get_user_details
list_all_airports
search_direct_flight
search_onestop_flight |
| **Write APIs** | cancel_pending_order
exchange_delivered_order_items
modify_pending_order_address
modify_pending_order_items
modify_pending_order_payment
modify_user_address
return_delivered_order_items | book_reservation
cancel_reservation
send_certificate
update_reservation_baggages
update_reservation_flights
update_reservation_passengers |
| **Non-DB APIs** | calculate, transfer_to_human_agents | |
| **Policies** | See B.1 | See B.1 |

Table 5: Overview of $\tau$-retail and $\tau$-airline databases and APIs.

**API design example**  Here is the Python implementation of an API in $\tau$-retail.

```python
import json
from typing import Any, Dict, List

def exchange_delivered_order_items(
    data: Dict[str, Any],
    order_id: str,
    item_ids: List[str],
    new_item_ids: List[str],
    payment_method_id: str,
) -> str:
    products, orders, users = data["products"], data["orders"],
    ↪  data["users"]

    # check order exists and is delivered
    if order_id not in orders:
        return "Error: order not found"
    order = orders[order_id]
    if order["status"] != "delivered":
        return "Error: non-delivered order cannot be exchanged"

    # check the items to be exchanged exist
    all_item_ids = [item["item_id"] for item in order["items"]]
    for item_id in item_ids:
        if item_ids.count(item_id) > all_item_ids.count(item_id):
            return f"Error: {item_id} not found"

    # check new items exist and match old items and are available
    if len(item_ids) != len(new_item_ids):
        return "Error: the number of items to be exchanged should match"

    diff_price = 0
    for item_id, new_item_id in zip(item_ids, new_item_ids):
        item = [item for item in order["items"] if item["item_id"] ==
        ↪  item_id][0]
        product_id = item["product_id"]
```

```python
        if not (
            new_item_id in products[product_id]["variants"]
            and
            ↪   products[product_id]["variants"][new_item_id]["available"]
        ):
            return f"Error: new item {new_item_id} not found or
            ↪   available"

        old_price = item["price"]
        new_price =
        ↪   products[product_id]["variants"][new_item_id]["price"]
        diff_price += new_price - old_price

    diff_price = round(diff_price, 2)

    # check payment method exists and can cover the price difference if
    ↪   gift card
    if payment_method_id not in
    ↪   users[order["user_id"]]["payment_methods"]:
        return "Error: payment method not found"

    payment_method =
    ↪   users[order["user_id"]]["payment_methods"][payment_method_id]
    if payment_method["source"] == "gift_card" and
    ↪   payment_method["balance"] < diff_price:
        return "Error: insufficient gift card balance to pay for the
        ↪   price difference"

    # modify the order
    order["status"] = "exchange requested"
    order["exchange_items"] = sorted(item_ids)
    order["exchange_new_items"] = sorted(new_item_ids)
    order["exchange_payment_method_id"] = payment_method_id
    order["exchange_price_difference"] = diff_price

    return json.dumps(order)

exchange_delivered_order_items.__info__ = {
    "type": "function",
    "function": {
        "name": "exchange_delivered_order_items",
        "description": "Exchange items in a delivered order to new items
        ↪   of the same product type. For a delivered order, return or
        ↪   exchange can be only done once by the agent. The agent needs
        ↪   to explain the exchange detail and ask for explicit user
        ↪   confirmation (yes/no) to proceed.",
        "parameters": {
            "type": "object",
            "properties": {
                "order_id": {
                    "type": "string",
                    "description": "The order id, such as '#W0000000'. Be
                    ↪   careful there is a '#' symbol at the beginning of
                    ↪   the order id.",
                },
                "item_ids": {
                    "type": "array",
                    "items": {
                        "type": "string",
                    },
                    "description": "The item ids to be exchanged, each
                    ↪   such as '1008292230'. There could be duplicate
                    ↪   items in the list.",
                },
```

```
                    "new_item_ids": {
                        "type": "array",
                        "items": {
                            "type": "string",
                        },
                        "description": "The item ids to be exchanged for,
                        ↪   each such as '1008292230'. There could be
                        ↪   duplicate items in the list. Each new item id
                        ↪   should match the item id in the same position and
                        ↪   be of the same product.",
                    },
                    "payment_method_id": {
                        "type": "string",
                        "description": "The payment method id to pay or
                        ↪   receive refund for the item price difference,
                        ↪   such as 'gift_card_0000000' or
                        ↪   'credit_card_0000000'. These can be looked up
                        ↪   from the user or order details.",
                    },
                },
                "required": [
                    "order_id",
                    "item_ids",
                    "new_item_ids",
                    "payment_method_id",
                ],
            },
        },
}
```

**Retail policies**

# Retail agent policy

```
As a retail agent, you can help users cancel or modify pending orders,
return or exchange delivered orders, modify their default user address,
or provide information about their own profile, orders, and related
products.

- At the beginning of the conversation, you have to authenticate the user
identity by locating their user id via email, or via name + zip code.
This has to be done even when the user already provides the user id.

- Once the user has been authenticated, you can provide the user with
information about order, product, profile information, e.g. help the user
look up order id.

- You can only help one user per conversation (but you can handle
multiple requests from the same user), and must deny any requests for
tasks related to any other user.

- Before taking consequential actions that update the database (cancel,
modify, return, exchange), you have to list the action detail and obtain
explicit user confirmation (yes) to proceed.

- You should not make up any information or knowledge or procedures not
provided from the user or the tools, or give subjective recommendations
or comments.

- You should at most make one tool call at a time, and if you take a tool
call, you should not respond to the user at the same time. If you respond
to the user, you should not make a tool call.

- You should transfer the user to a human agent if and only if the
request cannot be handled within the scope of your actions.
```

## Domain basic

– All times in the database are EST and 24 hour based. For example "02:30:00" means 2:30 AM EST.

– Each user has a profile of its email, default address, user id, and payment methods. Each payment method is either a gift card, a paypal account, or a credit card.

– Our retail store has 50 types of products. For each type of product, there are variant items of different options. For example, for a 't shirt' product, there could be an item with option 'color blue size M', and another item with option 'color red size L'.

– Each product has an unique product id, and each item has an unique item id. They have no relations and should not be confused.

– Each order can be in status 'pending', 'processed', 'delivered', or 'cancelled'. Generally, you can only take action on pending or delivered orders.

– Exchange or modify order tools can only be called once. Be sure that all items to be changed are collected into a list before making the tool call!!!

## Cancel pending order

– An order can only be cancelled if its status is 'pending', and you should check its status before taking the action.

– The user needs to confirm the order id and the reason (either 'no longer needed' or 'ordered by mistake') for cancellation.

– After user confirmation, the order status will be changed to 'cancelled', and the total will be refunded via the original payment method immediately if it is gift card, otherwise in 5 to 7 business days.

## Modify pending order

– An order can only be modified if its status is 'pending', and you should check its status before taking the action.

– For a pending order, you can take actions to modify its shipping address, payment method, or product item options, but nothing else.

### Modify payment

– The user can only choose a single payment method different from the original payment method.

– If the user wants the modify the payment method to gift card, it must have enough balance to cover the total amount.

– After user confirmation, the order status will be kept 'pending'. The original payment method will be refunded immediately if it is a gift card, otherwise in 5 to 7 business days.

### Modify items

– This action can only be called once, and will change the order status to 'pending (items modifed)', and the agent will not be able to modify or cancel the order anymore. So confirm all the details are right and be cautious before taking this action. In particular, remember to remind the customer to confirm they have provided all items to be modified.

– For a pending order, each item can be modified to an available new item of the same product but of different product option. There cannot be any change of product types, e.g. modify shirt to shoe.

– The user must provide a payment method to pay or receive refund of the price difference. If the user provides a gift card, it must have enough balance to cover the price difference.

## Return delivered order

– An order can only be returned if its status is 'delivered', and you should check its status before taking the action.

– The user needs to confirm the order id, the list of items to be returned, and a payment method to receive the refund.

– The refund must either go to the original payment method, or an existing gift card.

– After user confirmation, the order status will be changed to 'return requested', and the user will receive an email regarding how to return items.

## Exchange delivered order

– An order can only be exchanged if its status is 'delivered', and you should check its status before taking the action. In particular, remember to remind the customer to confirm they have provided all items to be exchanged.

– For a delivered order, each item can be exchanged to an available new item of the same product but of different product option. There cannot be any change of product types, e.g. modify shirt to shoe.

– The user must provide a payment method to pay or receive refund of the price difference. If the user provides a gift card, it must have enough balance to cover the price difference.

– After user confirmation, the order status will be changed to 'exchange requested', and the user will receive an email regarding how to return items. There is no need to place a new order.

Airline policies

# Airline Agent Policy

The current time is 2024-05-15 15:00:00 EST.

As an airline agent, you can help users book, modify, or cancel flight reservations.

–   Before taking any actions that update the booking database (booking, modifying flights, editing baggage, upgrading cabin class, or updating passenger information), you must list the action details and obtain explicit user confirmation (yes) to proceed.

–   You should not provide any information, knowledge, or procedures not provided by the user or available tools, or give subjective recommendations or comments.

–   You should only make one tool call at a time, and if you make a tool call, you should not respond to the user simultaneously. If you respond to the user, you should not make a tool call at the same time.

- You should deny user requests that are against this policy.

- You should transfer the user to a human agent if and only if the request cannot be handled within the scope of your actions.

## Domain Basic

- Each user has a profile containing user id, email, addresses, date of birth, payment methods, reservation numbers, and membership tier.

- Each reservation has an reservation id, user id, trip type (one way, round trip), flights, passengers, payment methods, created time, baggages, and travel insurance information.

- Each flight has a flight number, an origin, destination, scheduled departure and arrival time (local time), and for each date:
    - If the status is "available", the flight has not taken off, available seats and prices are listed.
    - If the status is "delayed" or "on time", the flight has not taken off, cannot be booked.
    - If the status is "flying", the flight has taken off but not landed, cannot be booked.

## Book flight

- The agent must first obtain the user id, then ask for the trip type, origin, destination.

- Passengers: Each reservation can have at most five passengers. The agent needs to collect the first name, last name, and date of birth for each passenger. All passengers must fly the same flights in the same cabin.

- Payment: each reservation can use at most one travel certificate, at most one credit card, and at most three gift cards. The remaining amount of a travel certificate is not refundable. All payment methods must already be in user profile for safety reasons.

- Checked bag allowance: If the booking user is a regular member, 0 free checked bag for each basic economy passenger, 1 free checked bag for each economy passenger, and 2 free checked bags for each business passenger. If the booking user is a silver member, 1 free checked bag for each basic economy passenger, 2 free checked bag for each economy passenger, and 3 free checked bags for each business passenger. If the booking user is a gold member, 2 free checked bag for each basic economy passenger, 3 free checked bag for each economy passenger, and 3 free checked bags for each business passenger. Each extra baggage is 50 dollars.

- Travel insurance: the agent should ask if the user wants to buy the travel insurance, which is 30 dollars per passenger and enables full refund if the user needs to cancel the flight given health or weather reasons.

## Modify flight

- The agent must first obtain the user id and the reservation id.

- Change flights: Basic economy flights cannot be modified. Other reservations can be modified without changing the origin, destination, and trip type. Some flight segments can be kept, but their prices will not be updated based on the current price. The API does not check these for the agent, so the agent must make sure the rules apply before calling the API!

-    Change cabin: all reservations, including basic economy, can change cabin without changing the flights. Cabin changes require the user to pay for the difference between their current cabin and the new cabin class. Cabin class must be the same across all the flights in the same reservation; changing cabin for just one flight segment is not possible.

-    Change baggage and insurance: The user can add but not remove checked bags. The user cannot add insurance after initial booking.

-    Change passengers: The user can modify passengers but cannot modify the number of passengers. This is something that even a human agent cannot assist with.

-    Payment: If the flights are changed, the user needs to provide one gift card or credit card for payment or refund method. The agent should ask for the payment or refund method instead.

## Cancel flight

-    The agent must first obtain the user id, the reservation id, and the reason for cancellation (change of plan, airline cancelled flight, or other reasons)

-    All reservations can be cancelled within 24 hours of booking, or if the airline cancelled the flight. Otherwise, basic economy or economy flights can be cancelled only if travel insurance is bought and the condition is met, and business flights can always be cancelled. The rules are strict regardless of the membership status. The API does not check these for the agent, so the agent must make sure the rules apply before calling the API!

-    The agent can only cancel the whole trip that is not flown. If any of the segments are already used, the agent cannot help and transfer is needed.

-    The refund will go to original payment methods in 5 to 7 business days.

## Refund

-    If the user is silver/gold member or has travel insurance or flies business, and complains about cancelled flights in a reservation, the agent can offer a certificate as a gesture after confirming the facts, with the amount being $100 times the number of passengers.

-    If the user is silver/gold member or has travel insurance or flies business, and complains about delayed flights in a reservation and wants to change or cancel the reservation, the agent can offer a certificate as a gesture after confirming the facts and changing or cancelling the reservation, with the amount being $50 times the number of passengers.

-    Do not proactively offer these unless the user complains about the situation and explicitly asks for some compensation. Do not compensate if the user is regular member and has no travel insurance and flies (basic) economy.

## B.2 STAGE II: DATA GENERATION

Below is the Python code to generate the `users` database in $\tau$-retail. As mentioned in the paper, the code is mostly generated by gpt-4 with some minor human editing. More data generation code can be seen in `https://github.com/sierra-research/tau-bench`.

We execute the code and use gpt-4 to refine the code based on execution for a few initial iterations, and manually refine the code to polish minor details. The data generation code uses `random` package to directly sample numeric and categorical entries like dates, prices, flight cabins, and sample from LM-generated lists for textual entries like product types or last names. This approach allows scalable data generation (i.e., we can sample 10,000 users if needed) with minimal human efforts.

```python
import random
import json
import numpy as np

# Updated cities list with corresponding zip code ranges for realism
cities_info = {
    "New York": ("NY", [10001, 10292]),
    "Los Angeles": ("CA", [90001, 91607]),
    "Chicago": ("IL", [60601, 60657]),
    "Houston": ("TX", [77001, 77299]),
    "Phoenix": ("AZ", [85001, 85099]),
    "Philadelphia": ("PA", [19019, 19190]),
    "San Antonio": ("TX", [78201, 78299]),
    "San Diego": ("CA", [92101, 92199]),
    "Dallas": ("TX", [75201, 75398]),
    "San Jose": ("CA", [95101, 95196]),
    "Austin": ("TX", [78701, 78799]),
    "Jacksonville": ("FL", [32099, 32290]),
    "Fort Worth": ("TX", [76101, 76199]),
    "Columbus": ("OH", [43085, 43299]),
    "Charlotte": ("NC", [28201, 28299]),
    "San Francisco": ("CA", [94102, 94188]),
    "Indianapolis": ("IN", [46201, 46298]),
    "Seattle": ("WA", [98101, 98199]),
    "Denver": ("CO", [80201, 80299]),
    "Washington": ("DC", [20001, 20599])
}

first_names = ["Emma", "Liam", "Olivia", "Noah", "Ava", "Yusuf",
↪  "Isabella", "Lucas", "Mia", "Mason",
            "Sophia", "Ethan", "Aarav", "James", "Amelia", "Lei",
            ↪  "Harper", "Sofia", "Evelyn", "Mohamed",
            "Yara", "Raj", "Fatima", "Juan", "Daiki", "Mei", "Chen",
            ↪  "Ivan", "Anya", "Omar"]
last_names = ["Smith", "Johnson", "Patel", "Nguyen", "Garcia", "Silva",
↪  "Kim", "Santos", "Khan", "Li",
            "Kovacs", "Muller", "Rossi", "Hernandez", "Sanchez", "Ito",
            ↪  "Johansson", "Lopez", "Gonzalez", "Ahmed",
            "Brown", "Davis", "Wilson", "Anderson", "Thomas", "Taylor",
            ↪  "Moore", "Jackson", "Martin", "Lee"]

# Function to generate an address with a city-realistic zip code
def generate_address():
    streets = ["Maple Drive", "Oak Street", "Pine Lane", "Elm Street",
    ↪  "Cedar Avenue",
            "Hillcrest Drive", "Willow Lane", "Sunset Drive", "River
            ↪  Road", "Lakeview Drive",
            "Main Street", "Park Avenue", "Broadway", "Elm Avenue",
            ↪  "Highland Drive",
            "Chestnut Street", "Hickory Lane", "Spruce Street",
            ↪  "Cedar Street", "Laurel Lane"]
```

```python
        city, (state, zip_range) = random.choice(list(cities_info.items()))
        address1 = f"{random.randint(100, 999)} {random.choice(streets)}"
        address2 = f"Suite {random.randint(100, 999)}"
        zip_code = str(random.randint(zip_range[0], zip_range[1]))
        return {
                "address1": address1,
                "address2": address2,
                "city": city,
                "country": "USA",
                "province": state,
                "zip": zip_code
        }

def poisson_sample(lam, max_val):
    sample = 0
    # Use rejection sampling to ensure sample is capped at max_val
    while sample == 0 or sample >= max_val:
        sample = np.random.poisson(lam)
    return sample

def generate_payment_method_id(source):  # like
↪ "card_2423"/"paypal_2423"/"gift_card_0912"
    random_id = f"{source}_{random.randint(1000000, 9999999)}"
    existing_ids = [method["id"] for user in user_profiles.values() for
↪   method in user["payment_methods"].values()]
    if random_id not in existing_ids:
        return random_id
    else:
        return generate_payment_method_id(source)

# Generate payment methods
def payment_method():
    payment_types = ["credit_card", "paypal", "gift_card"]
    count_methods = poisson_sample(1, 5)
    payment_methods = []
    existing_methods = set()
    for _ in range(count_methods):
        payment_source = random.choice(payment_types)
        if payment_source == "credit_card":
            brand = random.choice(["visa", "mastercard"])
            if ("credit_card", brand) in existing_methods:
                continue
            payment_methods.append({
                "source": "credit_card",
                "brand": brand,
                "last_four": f"{random.randint(1000, 9999)}",
                "id": generate_payment_method_id("credit_card")
            })
            existing_methods.add(("credit_card", brand))
        elif payment_source == "paypal":
            if ("paypal",) in existing_methods:
                continue
            payment_methods.append({
                "source": "paypal",
                "id": generate_payment_method_id("paypal")
            })
            existing_methods.add(("paypal",))
        else:
            if ("gift_card",) in existing_methods:
                continue
            payment_methods.append({
                "source": "gift_card",
                "balance": random.randint(0, 100),
                "id": generate_payment_method_id("gift_card")
            })
```

```python
            existing_methods.add(("gift_card",))
    return payment_methods

# Generate user profiles with adjusted zip codes
user_profiles = {}
for i in range(500):
    first_name = random.choice(first_names)
    last_name = random.choice(last_names)
    email_suffix, user_id_suffix = random.sample(range(1000, 10000), 2)
    email =
    ↪  f"{first_name.lower()}.{last_name.lower()}{email_suffix}@example.com"
    user_id =
    ↪  f"{first_name.lower()}_{last_name.lower()}_{user_id_suffix}"
    # addresses = [generate_address() for _ in range(poisson_sample(1,
    ↪  3))]
    payment_methods = payment_method()

    user_profiles[user_id] = {
        "name": {
            "first_name": first_name,
            "last_name": last_name
        },
        "address": generate_address(),
        "email": email,
        "payment_methods": {method['id']: method for method in
        ↪  payment_methods},
    }
```

# C  RETAIL EXAMPLES

## C.1  DATA EXAMPLES

Here are some examples from `users/products/orders.json` respectively. All data is generated by code, and the code is mostly generated by gpt-4, and the gpt-4 prompt is generated by authors.

```json
{
    "name": { "first_name": "James", "last_name": "Li" },
    "address": {
        "address1": "215 River Road",
        "address2": "Suite 991",
        "city": "New York",
        "country": "USA",
        "province": "NY",
        "zip": "10083"
    },
    "email": "james.li4495@example.com",
    "payment_methods": {
        "gift_card_1725971": { "source": "gift_card", "balance": 17, "id": "gift_card_1725971"
    },
    "orders": ["#W2611340", "#W3632959", "#W4435622", "#W3638028"]
}
```

Listing 1: An example entry from users database in $\tau$-retail.

```json
{
    "name": "Office Chair",
    "product_id": "4794339885",
    "variants": {
        "1793929609": {
            "item_id": "1793929609",
            "options": {
                "material": "fabric",
                "color": "black",
                "armrest": "none",
                "backrest height": "high-back"
            },
            "available": true,
            "price": 514.34
        },
        "4274709903": {
            "item_id": "4274709903",
            "options": {
                "material": "mesh",
                "color": "red",
                "armrest": "none",
                "backrest height": "standard"
            },
            "available": true,
            "price": 544.29
        }, ...
    }
}
```

Listing 2: An example entry from products database in $\tau$-retail (more variants omitted).

```json
{
    "order_id": "#W2611340",
    "user_id": "james_li_5688",
    "address": {
        "address1": "215 River Road",
        "address2": "Suite 991",
        "city": "New York",
        "country": "USA",
        "state": "NY",
        "zip": "10083"
    },
    "items": [
        {
            "name": "Water Bottle",
            "product_id": "8310926033",
            "item_id": "6469567736",
            "price": 47.84,
            "options": {
                "capacity": "1000ml",
                "material": "glass",
                "color": "blue"
            }
        },
        {
            "name": "Office Chair",
            "product_id": "4794339885",
            "item_id": "8426249116",
            "price": 488.81,
            "options": {
                "material": "fabric",
                "color": "black",
                "armrest": "fixed",
                "backrest height": "standard"
            }
        }
    ],
    "fulfillments": [
        {
            "tracking_id": ["357962501027"],
            "item_ids": ["6469567736", "8426249116"]
        }
    ],
    "status": "processed",
    "payment_history": [
        {
            "transaction_type": "payment",
            "amount": 536.65,
            "payment_method_id": "gift_card_1725971"
        }
    ]
}
```

Listing 3: An example entry from orders database in $\tau$-retail.

### C.2 TASK AND TRAJECTORY EXAMPLES

Here, tasks are not cherry-picked, and the trajectories are based on gpt-4o function calling agent. **Note that the task annotation (in JSON) is hidden to the agent.**

#### C.2.1 TASK 0: WRONG DECISION

Here, the agent omits the policy that only one exchange per order is possible, and issues one item exchange first without informing the user about this policy or recalling such a policy itself.

```
{
  "instruction": "You are Yusuf Rossi in 19122. You received your order
  #W2378156 and wish to exchange the mechanical keyboard for a similar
  one but with clicky switches and the smart thermostat for one
  compatible with Google Home instead of Apple HomeKit. If there is no
  keyboard that is clicky, RGB backlight, full size, you'd go for no
  backlight. You are detail-oriented and want to make sure everything is
  addressed in one go.",
  "actions": [
    {
      "name": "exchange_delivered_order_items",
      "arguments": {
        "order_id": "#W2378156",
        "item_ids": [
          "1151293680",
          "4983901480"
        ],
        "new_item_ids": [
          "7706410293",
          "7747408585"
        ],
        "payment_method_id": "credit_card_9513926"
      }
    }
  ]
}
```

**user:** Hello there! I've just received my order with the number W2378156, and I'd like to inquire about making a couple of exchanges.

**assistant:** Sure, I can help you with that. First, I need to authenticate your identity. Could you please provide your email address?

**user:** Apologies, but I am not comfortable sharing my email in chat. However, I can confirm the name on the order is Yusuf Rossi and shipping zip code as 19122. Would that be sufficient?

**assistant:** Function(arguments='{"first_name":"Yusuf","last_name":"Rossi","z⌋ip":"19122"}',
name='find_user_id_by_name_zip')

**tool:** yusuf_rossi_9620

**assistant:** Thank you, Yusuf. I have successfully authenticated your identity.

Now, let's proceed with the exchange. Could you please provide the details of the items you want to exchange and the new items you want in return?

**user**: Absolutely. Starting with the mechanical keyboard from this order, I'd like to exchange it for a similar one but with clicky switches. It's also important that it has RGB backlighting and that it's a full-size model.

**assistant**: Function(arguments='{"order_id":"#W2378156"}', name='get_order_details')

**tool**: {"order_id": "#W2378156", "user_id": "yusuf_rossi_9620", "address": {"address1": "763 Broadway", "address2": "Suite 135", "city": "Philadelphia", "country": "USA", "state": "PA", "zip": "19122"}, "items": [{"name": "Headphones", "product_id": "6992792935", "item_id": "4202497723", "price": 342.81, "options": {"type": "over-ear", "connectivity": "wireless", "color": "blue"}}, {"name": "Vacuum Cleaner", "product_id": "1762337868", "item_id": "4602305039", "price": 561.05, "options": {"type": "robotic", "bagged/bagless": "bagged", "features": "cordless"}}, {"name": "Mechanical Keyboard", "product_id": "1656367028", "item_id": "1151293680", "price": 272.33, "options": {"switch type": "linear", "backlight": "RGB", "size": "full size"}}, {"name": "Smart Thermostat", "product_id": "4896585277", "item_id": "4983901480", "price": 262.47, "options": {"compatibility": "Apple HomeKit", "color": "black"}}, {"name": "Smart Watch", "product_id": "6945232052", "item_id": "9408160950", "price": 381.26, "options": {"color": "gold", "band material": "leather", "display": "LCD"}}], "fulfillments": [{"tracking_id": ["843053632392"], "item_ids": ["4202497723", "4602305039", "1151293680", "4983901480", "9408160950"]}], "status": "delivered", "payment_history": [{"transaction_type": "payment", "amount": 1819.92, "payment_method_id": "credit_card_9513926"}]}

**assistant**: Function(arguments='{"product_id":"1656367028"}', name='get_product_details')

**tool**: {"name": "Mechanical Keyboard", "product_id": "1656367028", "variants": {"9690244451": {"item_id": "9690244451", "options": {"switch type": "clicky", "backlight": "RGB", "size": "60%"}, "available": false, "price": 236.51}, "7706410293": {"item_id": "7706410293", "options": {"switch type": "clicky", "backlight": "none", "size": "full size"}, "available": true, "price": 269.16}, "3616838507": {"item_id": "3616838507", "options": {"switch type": "tactile", "backlight": "white", "size": "full size"}, "available": true, "price": 226.11}, "8484921793": {"item_id": "8484921793", "options": {"switch type": "linear", "backlight": "RGB", "size": "80%"}, "available": true, "price": 230.15}, "1340995114": {"item_id": "1340995114", "options": {"switch type": "tactile", "backlight": "none", "size": "full size"}, "available": false, "price": 235.13}, "6342039236": {"item_id": "6342039236", "options": {"switch type": "clicky", "backlight": "white", "size": "full size"}, "available": true, "price": 244.91}, "1421289881": {"item_id": "1421289881", "options": {"switch type": "linear", "backlight": "none", "size": "80%"}, "available": true, "price": 268.77}, "7867398203": {"item_id": "7867398203", "options": {"switch type": "linear", "backlight": "RGB", "size": "60%"}, "available": true, "price": 232.7}, "4648814700": {"item_id": "4648814700", "options": {"switch type": "linear", "backlight": "white", "size": "60%"}, "available": false, "price": 228.84}, "5222576926": {"item_id": "5222576926", "options": {"switch type": "linear", "backlight": "white", "size": "full size"}, "available": false, "price": 249.95}, "4402162122": {"item_id": "4402162122", "options": {"switch type": "tactile", "backlight": "RGB", "size": "60%"}, "available": true, "price": 233.9}, "1151293680": {"item_id": "1151293680", "options": {"switch type": "linear", "backlight": "RGB", "size": "full size"}, "available": true, "price": 272.33}, "2299424241": {"item_id": "2299424241", "options": {"switch type": "clicky", "backlight": "RGB", "size": "80%"}, "available": true, "price": 237.48}, "4843487907": {"item_id": "4843487907", "options": {"switch type": "clicky", "backlight": "white", "size": "80%"}, "available": false, "price": 254.84}, "9025753381": {"item_id": "9025753381", "options": {"switch type": "clicky", "backlight": "RGB", "size": "full size"}, "available": false, "price": 231.58}, "6439196450": {"item_id": "6439196450", "options": {"switch type": "tactile", "backlight": "none", "size": "60%"}, "available": false, "price": 254.56}, "9991484137": {"item_id": "9991484137", "options": {"switch type": "tactile", "backlight": "white", "size": "80%"}, "available": true, "price": 240.97}, "9665000388": {"item_id": "9665000388", "options": {"switch type": "clicky", "backlight": "none", "size": "80%"}, "available": true, "price": 269.46}, "9570044148": {"item_id": "9570044148", "options": {"switch type": "linear", "backlight": "none", "size": "full size"}, "available": true, "price": 231.37}, "7658724607": {"item_id": "7658724607", "options": {"switch type": "tactile", "backlight": "none", "size": "80%"}, "available": true, "price": 256.73}}}

**assistant**: Here are the available options for mechanical keyboards with clicky switches and RGB backlighting in a full-size model:

1. **Item ID: 7706410293**
   - Switch Type: Clicky
   - Backlight: None
   - Size: Full Size
   - Price: $269.16

2. **Item ID: 6342039236**
   - Switch Type: Clicky
   - Backlight: White
   - Size: Full Size
   - Price: $244.91

Unfortunately, there are no full-size models with clicky switches and RGB backlighting currently available. Would you like to proceed with one of the available options, or would you prefer to wait until the desired model is back in stock?

**user**: Understandable. Hmm, in that case, I think I'll prioritize the clicky switches and go with the full-size model with no backlight, that is, the Item ID: 7706410293.

**assistant**: Great choice! Here are the details for the exchange:

- **Current Item:** Mechanical Keyboard (Item ID: 1151293680)
  - Switch Type: Linear
  - Backlight: RGB
  - Size: Full Size
  - Price: $272.33

- **New Item:** Mechanical Keyboard (Item ID: 7706410293)
  - Switch Type: Clicky
  - Backlight: None
  - Size: Full Size
  - Price: $269.16

The price difference will be refunded to your original payment method, which is a credit card.

Please confirm if you would like to proceed with this exchange.

**user**: Yes, I can confirm that this exchange is satisfactory. The clicky switches are far more significant to me than the backlight. Let's proceed with this exchange.

**assistant**:
Function(arguments='{"order_id":"#W2378156","item_ids":["1151293680"],"new_item_ids":["7706410293"],"payment_method_id":"credit_card_9513926"}', name='exchange_delivered_order_items')

**tool**: {"order_id": "#W2378156", "user_id": "yusuf_rossi_9620", "address": {"address1": "763 Broadway", "address2": "Suite 135", "city": "Philadelphia", "country": "USA", "state": "PA", "zip": "19122"}, "items": [{"name": "Headphones", "product_id": "6992792935", "item_id": "4202497723", "price": 342.81, "options": {"type": "over-ear", "connectivity": "wireless", "color": "blue"}}, {"name": "Vacuum Cleaner", "product_id": "1762337868", "item_id": "4602305039", "price": 561.05, "options": {"type": "robotic", "bagged/bagless": "bagged", "features": "cordless"}}, {"name": "Mechanical Keyboard", "product_id": "1656367028", "item_id": "1151293680", "price": 272.33, "options": {"switch type": "linear", "backlight": "RGB", "size": "full size"}}, {"name": "Smart Thermostat", "product_id": "4896585277", "item_id": "4983901480", "price": 262.47, "options": {"compatibility": "Apple HomeKit", "color": "black"}}, {"name": "Smart Watch", "product_id": "6945232052", "item_id": "9408160950", "price": 381.26, "options": {"color": "gold", "band material": "leather", "display": "LCD"}}], "fulfillments": [{"tracking_id": ["843053632392"], "item_ids": ["4202497723", "4602305039", "1151293680", "4983901480", "9408160950"]}], "status": "exchange requested", "payment_history": [{"transaction_type": "payment", "amount": 1819.92, "payment_method_id": "credit_card_9513926"}], "exchange_items": ["1151293680"], "exchange_new_items": ["7706410293"], "exchange_payment_method_id": "credit_card_9513926", "exchange_price_difference": -3.17}

**assistant**: The exchange request has been successfully processed. Here are the details:

- **Current Item:** Mechanical Keyboard (Item ID: 1151293680)
  - Switch Type: Linear
  - Backlight: RGB
  - Size: Full Size
  - Price: $272.33

- **New Item:** Mechanical Keyboard (Item ID: 7706410293)
  - Switch Type: Clicky
  - Backlight: None
  - Size: Full Size
  - Price: $269.16

- **Price Difference:** $3.17 (will be refunded to your credit card)

You will receive an email with instructions on how to return the current item. Once the return is processed, the new item will be shipped to you.

Is there anything else I can assist you with today?

**user**: Yes, there's another item I'd like to discuss for exchange. Specifically, the smart thermostat from my order; it's currently compatible with Apple HomeKit, but I'd like to exchange it for one that's compatible with Google Home.

**assistant**: Function(arguments='{"product_id":"4896585277"}', name='get_product_details')

**tool**: {"name": "Smart Thermostat", "product_id": "4896585277", "variants": {"8722653925": {"item_id": "8722653925", "options": {"compatibility": "Google Assistant", "color": "white"}, "available": false, "price": 227.8}, "8593894906": {"item_id": "8593894906", "options": {"compatibility": "Amazon Alexa", "color": "white"}, "available": false, "price": 263.11}, "2791467853": {"item_id": "2791467853", "options": {"compatibility": "Google Assistant", "color": "stainless steel"}, "available": false, "price": 242.53}, "7747408585": {"item_id": "7747408585", "options": {"compatibility": "Google Assistant", "color": "black"}, "available": true, "price": 249.01}, "4953074738": {"item_id": "4953074738", "options": {"compatibility": "Amazon Alexa", "color": "black"}, "available": true, "price": 226.02}, "4983901480": {"item_id": "4983901480", "options": {"compatibility": "Apple HomeKit", "color": "black"}, "available": true, "price": 262.47}, "9480266227": {"item_id": "9480266227", "options": {"compatibility": "Apple HomeKit", "color": "stainless steel"}, "available": true, "price": 255.98}, "6243148452": {"item_id": "6243148452", "options": {"compatibility": "Amazon Alexa", "color": "stainless steel"}, "available": true, "price": 247.0}, "3377900078": {"item_id": "3377900078", "options": {"compatibility": "Apple HomeKit", "color": "white"}, "available": true, "price": 260.68}}}

**assistant**: Here are the available options for smart thermostats that are compatible with Google Assistant:

1. **Item ID: 7747408585**
   - Compatibility: Google Assistant
   - Color: Black
   - Price: $249.01

Unfortunately, this is the only available option that matches your requirement. Would you like to proceed with this exchange?

**user**: That item fits the particular requirement I had in mind. Yes, let's proceed with the exchange for the thermostat.

**assistant**: Here are the details for the exchange:

- **Current Item:** Smart Thermostat (Item ID: 4983901480)
  - Compatibility: Apple HomeKit
  - Color: Black
  - Price: $262.47

- **New Item:** Smart Thermostat (Item ID: 7747408585)
  - Compatibility: Google Assistant
  - Color: Black
  - Price: $249.01

The price difference will be refunded to your original payment method, which is a credit card.

Please confirm if you would like to proceed with this exchange.

**user**: Yes, I confirm that I would like to exchange the thermostat as well. The compatibility with Google Assistant is really crucial for me, so it seems like the right choice.

**assistant**:
Function(arguments='{"order_id":"#W2378156","item_ids":["4983901480"],"new_item_ids":["7747408585"],"payment_method_id":"credit_card_9513926"}', name='exchange_delivered_order_items')

**tool**: Error: non-delivered order cannot be exchanged

**assistant**: Function(arguments='{"summary":"User Yusuf Rossi wants to exchange a smart thermostat (Item ID: 4983901480) for a Google Assistant compatible model (Item ID: 7747408585). The system returned an error stating the order is not marked as delivered, but the order status is indeed \'delivered\'. Please assist with the exchange."}', name='transfer_to_human_agents')

**tool**: Transfer successful

### C.2.2 TASK 7: WRONG ARGUMENT

Here, the agent fails as it is not able to reason over complex product inventories to find the desired exchange item option.

```
{
  "instruction": "You are mei_kovacs_8020 (28236) and you want to
  exchange the water bottle and the desk lamp. You want to exchange the
  water bottle to a bigger one, and the desk lamp to a less bright one
  (prefer AC adapter > battery > USB). If the agent asks for
  confirmation, only exchange the desk lamp.",
  "actions": [
    {
      "name": "exchange_delivered_order_items",
      "arguments": {
        "order_id": "#W6390527",
        "item_ids": ["8384507844"],
        "new_item_ids": ["1569765161"],
        "payment_method_id": "paypal_7644869"
      }
    }
  ]
}
```

**user**: Hello! I bought a water bottle and a desk lamp from your store and I'd like to exchange them both. Could you assist me with it?

**assistant**: Sure, I can help you with that. First, I need to authenticate your identity. Could you please provide your email address?

**user**: Sure, it's mei_kovacs_8020@mail.com.

**assistant**: Function(arguments='{"email":"mei_kovacs_8020@mail.com"}', name='find_user_id_by_email')

**tool**: Error: user not found

**assistant**: It seems I couldn't find a user with that email address. Could you please provide your first name, last name, and zip code instead?

**user**: Oh, my apologies for the confusion. My name is Mei Kovacs and my zip code is 28236.

**assistant**: Function(arguments='{"first_name":"Mei","last_name":"Kovacs","zip":"28236"}', name='find_user_id_by_name_zip')

**tool**: mei_kovacs_8020

**assistant**: Function(arguments='{"user_id":"mei_kovacs_8020"}', name='get_user_details')

**tool**: {"name": {"first_name": "Mei", "last_name": "Kovacs"}, "address": {"address1": "317 Elm Street", "address2": "Suite 461", "city": "Charlotte", "country": "USA", "province": "NC", "zip": "28236"}, "email": "mei.kovacs8232@example.com", "payment_methods": {"paypal_7644869": {"source": "paypal", "id": "paypal_7644869"}}, "orders": ["#W6390527", "#W7800651", "#W8065207"]}

**assistant**: Thank you, Mei. I have successfully authenticated your identity. Now, I need to check the details of the orders containing the water bottle and desk lamp. Could you please provide the order ID for these items?

**user**: I'm afraid I don't remember the order ID as of now. I am unable to provide you with it, I'm sorry.

**assistant**: Function(arguments='{"order_id": "#W6390527"}', name='get_order_details')

**tool**: {"order_id": "#W6390527", "user_id": "mei_kovacs_8020", "address": {"address1": "317 Elm Street", "address2": "Suite 461", "city": "Charlotte", "country": "USA", "state": "NC", "zip": "28236"}, "items": [{"name": "Desk Lamp", "product_id": "6817146515", "item_id": "8384507844", "price": 137.94, "options": {"color": "white", "brightness": "medium", "power source": "USB"}}, {"name": "Hiking Boots", "product_id": "7363354090", "item_id": "1615379700", "price": 253.89, "options": {"size": "10", "material": "synthetic", "waterproof": "yes"}}, {"name": "Water Bottle", "product_id": "8310926033", "item_id": "8538875209", "price": 45.13, "options": {"capacity": "500ml", "material": "glass", "color": "black"}}], "fulfillments": [{"tracking_id": ["745984883162"], "item_ids": ["8384507844", "1615379700", "8538875209"]}], "status": "delivered", "payment_history": [{"transaction_type": "payment", "amount": 436.96, "payment_method_id": "paypal_7644869"}]}

**assistant**: I have found the order containing both the water bottle and the desk lamp. Here are the details:

- **Order ID:** #W6390527
- **Status:** Delivered
- **Items:**
  - Desk Lamp (Item ID: 8384507844)
  - Water Bottle (Item ID: 8538875209)

To proceed with the exchange, I need the details of the new items you would like to exchange these for. Please provide the item IDs of the new items.

**user**: I don't have the exact item IDs right now. However, could I tell you what I am looking for exactly instead?

**assistant**: Sure, you can describe the new items you want to exchange for, and I can look up the item IDs for you. Please provide the details of the new desk lamp and water bottle you are looking for.

**user:** Great! I would like to exchange my current water bottle for a larger one, and the desk lamp for one that is less bright. For the lamp, I would prefer one with an AC adapter over a battery-powered one, and definitely over a USB one, if possible.

**assistant:** Function(arguments='{"product_id": "8310926033"}', name='get_product_details')

**tool:** {"name": "Water Bottle", "product_id": "8310926033", "variants": {"1434748144": {"item_id": "1434748144", "options": {"capacity": "1000ml", "material": "glass", "color": "red"}, "available": false, "price": 49.72}, "4579334072": {"item_id": "4579334072", "options": {"capacity": "750ml", "material": "glass", "color": "black"}, "available": true, "price": 54.85}, "6469567736": {"item_id": "6469567736", "options": {"capacity": "1000ml", "material": "glass", "color": "blue"}, "available": false, "price": 47.84}, "3453331371": {"item_id": "3453331371", "options": {"capacity": "500ml", "material": "stainless steel", "color": "black"}, "available": true, "price": 52.79}, "2439754078": {"item_id": "2439754078", "options": {"capacity": "1000ml", "material": "stainless steel", "color": "red"}, "available": true, "price": 49.51}, "7843064651": {"item_id": "7843064651", "options": {"capacity": "750ml", "material": "stainless steel", "color": "blue"}, "available": true, "price": 50.14}, "7918497119": {"item_id": "7918497119", "options": {"capacity": "500ml", "material": "glass", "color": "blue"}, "available": false, "price": 54.51}, "5758737025": {"item_id": "5758737025", "options": {"capacity": "500ml", "material": "glass", "color": "green"}, "available": true, "price": 45.09}, "7533802601": {"item_id": "7533802601", "options": {"capacity": "500ml", "material": "stainless steel", "color": "green"}, "available": true, "price": 48.59}, "3229676465": {"item_id": "3229676465", "options": {"capacity": "500ml", "material": "plastic", "color": "black"}, "available": true, "price": 51.94}, "2366567022": {"item_id": "2366567022", "options": {"capacity": "1000ml", "material": "stainless steel", "color": "blue"}, "available": false, "price": 54.04}, "6974536207": {"item_id": "6974536207", "options": {"capacity": "750ml", "material": "plastic", "color": "blue"}, "available": true, "price": 49.3}, "6777246137": {"item_id": "6777246137", "options": {"capacity": "750ml", "material": "stainless steel", "color": "red"}, "available": true, "price": 47.76}, "8538875209": {"item_id": "8538875209", "options": {"capacity": "500ml", "material": "glass", "color": "black"}, "available": true, "price": 45.13}, "9127591879": {"item_id": "9127591879", "options": {"capacity": "750ml", "material": "stainless steel", "color": "black"}, "available": false, "price": 48.47}, "7661609223": {"item_id": "7661609223", "options": {"capacity": "1000ml", "material": "stainless steel", "color": "black"}, "available": true, "price": 46.51}, "4947921075": {"item_id": "4947921075", "options": {"capacity": "750ml", "material": "stainless steel", "color": "green"}, "available": false, "price": 49.57}, "7199146548": {"item_id": "7199146548", "options": {"capacity": "750ml", "material": "plastic", "color": "black"}, "available": true, "price": 48.02}}}

**assistant:** Function(arguments='{"product_id":"6817146515"}', name='get_product_details')

**tool**: {"name": "Desk Lamp", "product_id": "6817146515", "variants": {"9083642334": {"item_id": "9083642334", "options": {"color": "white", "brightness": "high", "power source": "USB"}, "available": true, "price": 164.28}, "4385534692": {"item_id": "4385534692", "options": {"color": "white", "brightness": "high", "power source": "AC adapter"}, "available": false, "price": 138.07}, "7624783998": {"item_id": "7624783998", "options": {"color": "black", "brightness": "high", "power source": "AC adapter"}, "available": true, "price": 154.17}, "1270145486": {"item_id": "1270145486", "options": {"color": "white", "brightness": "high", "power source": "battery"}, "available": false, "price": 144.07}, "5320792178": {"item_id": "5320792178", "options": {"color": "black", "brightness": "medium", "power source": "AC adapter"}, "available": true, "price": 135.24}, "5370728469": {"item_id": "5370728469", "options": {"color": "silver", "brightness": "medium", "power source": "USB"}, "available": true, "price": 164.97}, "6805564527": {"item_id": "6805564527", "options": {"color": "black", "brightness": "medium", "power source": "USB"}, "available": true, "price": 158.41}, "1569765161": {"item_id": "1569765161", "options": {"color": "silver", "brightness": "low", "power source": "AC adapter"}, "available": true, "price": 143.02}, "7453605304": {"item_id": "7453605304", "options": {"color": "silver", "brightness": "low", "power source": "battery"}, "available": true, "price": 150.01}, "9190635437": {"item_id": "9190635437", "options": {"color": "black", "brightness": "low", "power source": "USB"}, "available": true, "price": 153.23}, "4447749792": {"item_id": "4447749792", "options": {"color": "white", "brightness": "medium", "power source": "AC adapter"}, "available": false, "price": 139.8}, "8384507844": {"item_id": "8384507844", "options": {"color": "white", "brightness": "medium", "power source": "USB"}, "available": false, "price": 137.94}}}

**assistant**: Here are the available options for the water bottle and desk lamp that match your preferences:

### Water Bottle Options:
1. **Item ID:** 7661609223
   - Capacity: 1000ml
   - Material: Stainless Steel
   - Color: Black
   - Price: $46.51

2. **Item ID:** 2439754078
   - Capacity: 1000ml
   - Material: Stainless Steel
   - Color: Red
   - Price: $49.51

### Desk Lamp Options:
1. **Item ID:** 5320792178
   - Color: Black
   - Brightness: Medium
   - Power Source: AC Adapter
   - Price: $135.24

2. **Item ID:** 1569765161
   - Color: Silver
   - Brightness: Low
   - Power Source: AC Adapter
   - Price: $143.02

Please confirm which items you would like to exchange for, and provide the payment method you would like to use for any price difference.

**user**: I would like to exchange the desk lamp for the one with item ID 5320792178. As for the water bottle, I need some time to think about it. You can proceed with the desk lamp exchange for now.

**assistant**: Please note that for a delivered order, return or exchange can only be done once by the agent. If you proceed with the desk lamp exchange now, you won't be able to exchange the water bottle later through me.

Would you like to proceed with the desk lamp exchange now, or would you prefer to decide on both items before proceeding?

**user**: Alright, let's just proceed with the exchange for the desk lamp only. I understand the limitations, thank you for explaining it to me.

**assistant**: To proceed with the exchange, here are the details:

- **Current Desk Lamp:**
  - Item ID: 8384507844
  - Color: White
  - Brightness: Medium
  - Power Source: USB
  - Price: $137.94

- **New Desk Lamp:**
  - Item ID: 5320792178
  - Color: Black
  - Brightness: Medium
  - Power Source: AC Adapter
  - Price: $135.24

The price difference will be refunded to your PayPal account.

Please confirm if you would like to proceed with this exchange (yes/no).

**user**: Yes, I would like to proceed with this exchange. Thank you.

**assistant**:
```
Function(arguments='{"order_id":"#W6390527","item_ids":["8384507844"],"n↵
ew_item_ids":["5320792178"],"payment_method_id":"paypal_7644869"}',
name='exchange_delivered_order_items')
```

**tool**: {"order_id": "#W6390527", "user_id": "mei_kovacs_8020", "address": {"address1": "317 Elm Street", "address2": "Suite 461", "city": "Charlotte", "country": "USA", "state": "NC", "zip": "28236"}, "items": [{"name": "Desk Lamp", "product_id": "6817146515", "item_id": "8384507844", "price": 137.94, "options": {"color": "white", "brightness": "medium", "power source": "USB"}}, {"name": "Hiking Boots", "product_id": "7363354090", "item_id": "1615379700", "price": 253.89, "options": {"size": "10", "material": "synthetic", "waterproof": "yes"}}, {"name": "Water Bottle", "product_id": "8310926033", "item_id": "8538875209", "price": 45.13, "options": {"capacity": "500ml", "material": "glass", "color": "black"}}], "fulfillments": [{"tracking_id": ["745984883162"], "item_ids": ["8384507844", "1615379700", "8538875209"]}], "status": "exchange requested", "payment_history": [{"transaction_type": "payment", "amount": 436.96, "payment_method_id": "paypal_7644869"}], "exchange_items": ["8384507844"], "exchange_new_items": ["5320792178"], "exchange_payment_method_id": "paypal_7644869", "exchange_price_difference": -2.7}

**assistant**: The exchange request for your desk lamp has been successfully processed. You will receive an email with instructions on how to return the current desk lamp. The price difference of $2.70 will be refunded to your PayPal account.

If you need any further assistance, feel free to ask. Have a great day!

**user**: ###STOP###

### C.2.3 TASK 42: PARTIALLY SOLVE COMPOUND REQUESTS

Here, the task requires the agent to check all orders to fix wrong addresses. However, the agent only fixes the jigsaw order address as the user suggests.

```
{
  "instruction": "Your name is Mei Patel, and you live in 445 Maple
  Drive, Suite 394, Fort Worth, Texas, 76165. You just created your user
  id mei_patel_7272 and ordered some things, but realized you might have
  typed your address wrong. You want to check it, and potentially correct
  all order addresses and your user address. After this, you'd like to
  check the jigsaw you ordered, and if it's not shipped yet, you want to
  change it to the easiest jigsaw (easiest level, least pieces) because
  your kid is too young. By default you use PayPal. You are brief and
  your memory is not too good sometimes, but you are polite.",
  "actions": [
    {
      "name": "modify_pending_order_address",
      "arguments": {
        "order_id": "#W9583042",
        "address1": "445 Maple Drive",
        "address2": "Suite 394",
        "city": "Fort Worth",
        "state": "TX",
        "country": "USA",
        "zip": "76165"
      }
    },
    {
      "name": "modify_pending_order_address",
      "arguments": {
        "order_id": "#W4082615",
        "address1": "445 Maple Drive",
        "address2": "Suite 394",
        "city": "Fort Worth",
        "state": "TX",
        "country": "USA",
        "zip": "76165"
      }
    },
    {
      "name": "modify_user_address",
      "arguments": {
        "user_id": "mei_patel_7272",
        "address1": "445 Maple Drive",
        "address2": "Suite 394",
        "city": "Fort Worth",
        "state": "TX",
        "country": "USA",
        "zip": "76165"
      }
    },
    {
      "name": "modify_pending_order_items",
      "arguments": {
        "order_id": "#W4082615",
        "item_ids": ["9779102705"],
        "new_item_ids": ["1096508426"],
        "payment_method_id": "paypal_4768213"
      }
    }
  ],
}
```

**user:** I think I might have entered my address wrongly during my recent order. Could you please check that for me?

**assistant:** Sure, I can help with that. First, I need to authenticate your identity. Could you please provide your email address?

**user:** I'm afraid I don't quite remember which email I used to register. Can we proceed with my user id instead? It's mei_patel_7272.

**assistant:** I still need to authenticate your identity. Could you please provide your first name, last name, and zip code? This will help me locate your user id.

**user:** Of course, my full name is Mei Patel and my zip code is 76165.

**assistant:** Function(arguments='{"first_name":"Mei","last_name":"Patel","zip⌋
":"76165"}',
name='find_user_id_by_name_zip')

**tool:** mei_patel_7272

**assistant:** Thank you for the information. Your user id has been authenticated successfully.

Could you please provide the order id for the order you want me to check the address for?

**user:** I don't recall the exact order id, is there a way to find my recent orders using my user id?

**assistant:** Function(arguments='{"user_id":"mei_patel_7272"}',
name='get_user_details')

**tool:** {"name": {"first_name": "Mei", "last_name": "Patel"}, "address": {"address1": "443 Maple Drive", "address2": "Suite 394", "city": "Fort Worth", "country": "USA", "province": "TX", "zip": "76165"}, "email": "mei.patel3193@example.com", "payment_methods": {"credit_card_9503061": {"source": "credit_card", "brand": "visa", "last_four": "9904", "id": "credit_card_9503061"}, "paypal_4768213": {"source": "paypal", "id": "paypal_4768213"}}, "orders": ["#W9583042", "#W4082615"]}

**assistant:** I found your recent orders. Here are the details:

1. Order ID: #W9583042
2. Order ID: #W4082615

Please let me know which order you would like me to check the address for.

**user:** I can't remember exactly which one, but it's the jigsaw that I ordered recently. Could we look into that one, please?

**assistant**: Function(arguments='{"order_id": "#W9583042"}', name='get_order_details')

**tool**: {"order_id": "#W9583042", "user_id": "mei_patel_7272", "address": {"address1": "443 Maple Drive", "address2": "Suite 394", "city": "Fort Worth", "country": "USA", "state": "TX", "zip": "76165"}, "items": [{"name": "Perfume", "product_id": "6858788497", "item_id": "5421902839", "price": 328.25, "options": {"scent family": "oriental", "size": "100ml", "gender": "men"}}, {"name": "Dumbbell Set", "product_id": "7233192239", "item_id": "6245231688", "price": 522.03, "options": {"weight range": "30-50 lbs", "material": "iron", "set type": "adjustable"}}, {"name": "Luggage Set", "product_id": "5426915165", "item_id": "7160999700", "price": 499.29, "options": {"piece count": "2-piece", "color": "red", "material": "softshell"}}, {"name": "Garden Hose", "product_id": "6679515468", "item_id": "3230708338", "price": 99.51, "options": {"length": "25ft", "material": "latex", "color": "green"}}], "fulfillments": [], "status": "pending", "payment_history": [{"transaction_type": "payment", "amount": 1449.08, "payment_method_id": "paypal_4768213"}]}

**assistant**: Function(arguments='{"order_id":"#W4082615"}', name='get_order_details')

**tool**: {"order_id": "#W4082615", "user_id": "mei_patel_7272", "address": {"address1": "443 Maple Drive", "address2": "Suite 394", "city": "Fort Worth", "country": "USA", "state": "TX", "zip": "76165"}, "items": [{"name": "Jigsaw Puzzle", "product_id": "1808611083", "item_id": "9779102705", "price": 54.11, "options": {"pieces": "1000", "theme": "art", "difficulty level": "intermediate"}}, {"name": "Backpack", "product_id": "2524789262", "item_id": "5917587651", "price": 212.79, "options": {"color": "grey", "size": "medium", "material": "polyester", "compartment": "laptop"}}, {"name": "Grill", "product_id": "6819683148", "item_id": "3876764226", "price": 981.47, "options": {"type": "electric", "size": "portable", "features": "side burner"}}, {"name": "Perfume", "product_id": "6858788497", "item_id": "8316205423", "price": 288.75, "options": {"scent family": "woody", "size": "30ml", "gender": "women"}}, {"name": "Smart Watch", "product_id": "6945232052", "item_id": "2540052208", "price": 346.42, "options": {"color": "gold", "band material": "silicone", "display": "LCD"}}], "fulfillments": [], "status": "pending", "payment_history": [{"transaction_type": "payment", "amount": 1883.54, "payment_method_id": "paypal_4768213"}]}

**assistant**: The order with the jigsaw puzzle is Order ID: #W4082615. The current shipping address for this order is:

- **Address Line 1:** 443 Maple Drive
- **Address Line 2:** Suite 394
- **City:** Fort Worth
- **State:** TX
- **Country:** USA
- **Zip Code:** 76165

Is this the address you intended to use? If not, please provide the correct address, and I can update it for you.

**user**: That's not right, my correct address should be 445 Maple Drive, Suite 394, Fort Worth, Texas, 76165. Could you please correct my user address and the order address to this?

**assistant:** I can update both your default user address and the shipping address for the pending order #W4082615 to the correct address:

- **Address Line 1:** 445 Maple Drive
- **Address Line 2:** Suite 394
- **City:** Fort Worth
- **State:** TX
- **Country:** USA
- **Zip Code:** 76165

Please confirm if this is correct, and I will proceed with the updates.

**user:** Yes, that's correct. Please go ahead and update them.

**assistant:** Function(arguments='{"user_id": "mei_patel_7272", "address1": "445 Maple Drive", "address2": "Suite 394", "city": "Fort Worth", "state": "TX", "country": "USA", "zip": "76165"}', name='modify_user_address')

**tool:** {"name": {"first_name": "Mei", "last_name": "Patel"}, "address": {"address1": "445 Maple Drive", "address2": "Suite 394", "city": "Fort Worth", "state": "TX", "country": "USA", "zip": "76165"}, "email": "mei.patel3193@example.com", "payment_methods": {"credit_card_9503061": {"source": "credit_card", "brand": "visa", "last_four": "9904", "id": "credit_card_9503061"}, "paypal_4768213": {"source": "paypal", "id": "paypal_4768213"}}, "orders": ["#W9583042", "#W4082615"]}

**assistant:** Function(arguments='{"order_id":"#W4082615","address1":"445 Maple Drive","address2":"Suite 394","city":"Fort Worth","state":"TX","country":"USA","zip":"76165"}', name='modify_pending_order_address')

**tool:** {"order_id": "#W4082615", "user_id": "mei_patel_7272", "address": {"address1": "445 Maple Drive", "address2": "Suite 394", "city": "Fort Worth", "state": "TX", "country": "USA", "zip": "76165"}, "items": [{"name": "Jigsaw Puzzle", "product_id": "1808611083", "item_id": "9779102705", "price": 54.11, "options": {"pieces": "1000", "theme": "art", "difficulty level": "intermediate"}}, {"name": "Backpack", "product_id": "2524789262", "item_id": "5917587651", "price": 212.79, "options": {"color": "grey", "size": "medium", "material": "polyester", "compartment": "laptop"}}, {"name": "Grill", "product_id": "6819683148", "item_id": "3876764226", "price": 981.47, "options": {"type": "electric", "size": "portable", "features": "side burner"}}, {"name": "Perfume", "product_id": "6858788497", "item_id": "8316205423", "price": 288.75, "options": {"scent family": "woody", "size": "30ml", "gender": "women"}}, {"name": "Smart Watch", "product_id": "6945232052", "item_id": "2540052208", "price": 346.42, "options": {"color": "gold", "band material": "silicone", "display": "LCD"}}], "fulfillments": [], "status": "pending", "payment_history": [{"transaction_type": "payment", "amount": 1883.54, "payment_method_id": "paypal_4768213"}]}

**assistant:** Both your default user address and the shipping address for the pending order #W4082615 have been successfully updated to:

- **Address Line 1:** 445 Maple Drive
- **Address Line 2:** Suite 394
- **City:** Fort Worth
- **State:** TX
- **Country:** USA

– **Zip Code:** 76165

Is there anything else I can assist you with?

**user:** Yes, there's actually one more thing. Could you check the jigsaw that I ordered, and let me know whether it has been shipped yet?

**assistant:** The order status for #W4082615 is currently "pending," which means it has not been processed or shipped yet.

Is there anything else you would like to know or need assistance with?

**user:** Yes, I'm considering changing the jigsaw I ordered. Could you tell me what is the easiest jigsaw available, like one with the least pieces fit for a young kid?

**assistant:** Function(arguments='{}', name='list_all_product_types')

**tool:** {"Action Camera": "3377618313", "Air Purifier": "3821016478", "Backpack": "2524789262", "Bicycle": "9783735446", "Bluetooth Speaker": "4768869376", "Bookshelf": "8600330539", "Coffee Maker": "7996920482", "Cycling Helmet": "7765186836", "Desk Lamp": "6817146515", "Digital Camera": "8940227892", "Dumbbell Set": "7233192239", "E-Reader": "3801771308", "Electric Kettle": "1075968781", "Electric Toothbrush": "7352963235", "Espresso Machine": "4354588079", "Fleece Jacket": "8560156827", "Gaming Mouse": "5713490933", "Garden Hose": "6679515468", "Grill": "6819683148", "Headphones": "6992792935", "Hiking Boots": "7363354090", "Indoor Security Camera": "2985987096", "Jigsaw Puzzle": "1808611083", "LED Light Bulb": "2696197613", "Laptop": "4760268021", "Luggage Set": "5426915165", "Makeup Kit": "5149340237", "Mechanical Keyboard": "1656367028", "Notebook": "2892623495", "Office Chair": "4794339885", "Patio Umbrella": "9743693396", "Perfume": "6858788497", "Pet Bed": "2747247837", "Portable Charger": "6942297802", "Running Shoes": "6938111410", "Skateboard": "1968349452", "Smart Thermostat": "4896585277", "Smart Watch": "6945232052", "Smartphone": "1801728040", "Sneakers": "7471004230", "Sunglasses": "7314138884", "T-Shirt": "9523456873", "Tablet": "8024098596", "Tea Kettle": "9832717871", "Vacuum Cleaner": "1762337868", "Wall Clock": "2344688344", "Water Bottle": "8310926033", "Wireless Earbuds": "9924732112", "Wristwatch": "6066914160", "Yoga Mat": "4635925001"}

**assistant:** Function(arguments='{"product_id":"1808611083"}', name='get_product_details')

**tool**: {"name": "Jigsaw Puzzle", "product_id": "1808611083", "variants": {"3614853563": {"item_id": "3614853563", "options": {"pieces": "2000", "theme": "art", "difficulty level": "intermediate"}, "available": false, "price": 46.99}, "4772738468": {"item_id": "4772738468", "options": {"pieces": "1000", "theme": "animals", "difficulty level": "beginner"}, "available": false, "price": 53.91}, "4068787148": {"item_id": "4068787148", "options": {"pieces": "500", "theme": "art", "difficulty level": "intermediate"}, "available": true, "price": 52.01}, "3112842858": {"item_id": "3112842858", "options": {"pieces": "1000", "theme": "fantasy", "difficulty level": "intermediate"}, "available": true, "price": 49.1}, "7869640094": {"item_id": "7869640094", "options": {"pieces": "2000", "theme": "animals", "difficulty level": "expert"}, "available": false, "price": 47.59}, "1096508426": {"item_id": "1096508426", "options": {"pieces": "500", "theme": "art", "difficulty level": "beginner"}, "available": true, "price": 46.13}, "9237024510": {"item_id": "9237024510", "options": {"pieces": "500", "theme": "animals", "difficulty level": "expert"}, "available": true, "price": 53.53}, "5546244844": {"item_id": "5546244844", "options": {"pieces": "1500", "theme": "art", "difficulty level": "intermediate"}, "available": true, "price": 51.59}, "1008948180": {"item_id": "1008948180", "options": {"pieces": "1000", "theme": "art", "difficulty level": "beginner"}, "available": false, "price": 54.34}, "6245746168": {"item_id": "6245746168", "options": {"pieces": "1500", "theme": "animals", "difficulty level": "intermediate"}, "available": true, "price": 46.0}, "7127170374": {"item_id": "7127170374", "options": {"pieces": "2000", "theme": "fantasy", "difficulty level": "beginner"}, "available": false, "price": 52.03}, "9370300555": {"item_id": "9370300555", "options": {"pieces": "1000", "theme": "art", "difficulty level": "expert"}, "available": false, "price": 45.9}, "9665100170": {"item_id": "9665100170", "options": {"pieces": "1500", "theme": "animals", "difficulty level": "beginner"}, "available": true, "price": 45.39}, "4572024853": {"item_id": "4572024853", "options": {"pieces": "1000", "theme": "animals", "difficulty level": "expert"}, "available": true, "price": 53.72}, "5645314103": {"item_id": "5645314103", "options": {"pieces": "2000", "theme": "animals", "difficulty level": "intermediate"}, "available": true, "price": 46.19}, "9779102705": {"item_id": "9779102705", "options": {"pieces": "1000", "theme": "art", "difficulty level": "intermediate"}, "available": false, "price": 54.11}, "9030221155": {"item_id": "9030221155", "options": {"pieces": "2000", "theme": "art", "difficulty level": "beginner"}, "available": false, "price": 51.98}, "5172162216": {"item_id": "5172162216", "options": {"pieces": "2000", "theme": "landscape", "difficulty level": "intermediate"}, "available": false, "price": 48.51}}}

**assistant**: The easiest jigsaw puzzle available, suitable for a young kid, is the one with the least pieces. Here are the options with 500 pieces:

1. **Item ID: 1096508426**
   - **Theme:** Art
   - **Difficulty Level:** Beginner
   - **Price:** $46.13

2. **Item ID: 9237024510**
   - **Theme:** Animals
   - **Difficulty Level:** Expert
   - **Price:** $53.53

Would you like to exchange your current jigsaw puzzle for one of these? If so, please confirm which one and provide a payment method to handle any price difference.

**user**: Could we go with the Art themed one, Item ID: 1096508426? I generally use PayPal for my purchases.

**assistant**: To proceed with the exchange, I will change the current jigsaw puzzle (Item ID: 9779102705) to the new one (Item ID: 1096508426) and use your PayPal account for any price difference.

Here are the details of the exchange:
- **Current Item:** Jigsaw Puzzle (1000 pieces, Art theme, Intermediate difficulty)
- **New Item:** Jigsaw Puzzle (500 pieces, Art theme, Beginner difficulty)
- **Payment Method:** PayPal (ID: paypal_4768213)

Please confirm if this is correct, and I will proceed with the exchange.

**user**: Yes, everything looks correct. Please go ahead and make the exchange.

**assistant**:
Function(arguments='{"order_id":"#W4082615","item_ids":["9779102705"],"new_item_ids":["1096508426"],"payment_method_id":"paypal_4768213"}', name='modify_pending_order_items')

**tool**: {"order_id": "#W4082615", "user_id": "mei_patel_7272", "address": {"address1": "445 Maple Drive", "address2": "Suite 394", "city": "Fort Worth", "state": "TX", "country": "USA", "zip": "76165"}, "items": [{"name": "Jigsaw Puzzle", "product_id": "1808611083", "item_id": "1096508426", "price": 46.13, "options": {"pieces": "500", "theme": "art", "difficulty level": "beginner"}}, {"name": "Backpack", "product_id": "2524789262", "item_id": "5917587651", "price": 212.79, "options": {"color": "grey", "size": "medium", "material": "polyester", "compartment": "laptop"}}, {"name": "Grill", "product_id": "6819683148", "item_id": "3876764226", "price": 981.47, "options": {"type": "electric", "size": "portable", "features": "side burner"}}, {"name": "Perfume", "product_id": "6858788497", "item_id": "8316205423", "price": 288.75, "options": {"scent family": "woody", "size": "30ml", "gender": "women"}}, {"name": "Smart Watch", "product_id": "6945232052", "item_id": "2540052208", "price": 346.42, "options": {"color": "gold", "band material": "silicone", "display": "LCD"}}], "fulfillments": [], "status": "pending (item modified)", "payment_history": [{"transaction_type": "payment", "amount": 1883.54, "payment_method_id": "paypal_4768213"}, {"transaction_type": "refund", "amount": 7.979999999999997, "payment_method_id": "paypal_4768213"}]}

**assistant**: The exchange has been successfully completed. Your order #W4082615 now includes the new jigsaw puzzle:

- **New Item:** Jigsaw Puzzle (500 pieces, Art theme, Beginner difficulty)

The payment difference has been handled using your PayPal account, and a refund of $7.98 has been processed.

Is there anything else I can assist you with?

**user**: No, that would be all. Thank you for your help.

**assistant**: You're welcome! If you have any more questions or need assistance in the future, feel free to reach out. Have a great day!

**user:** ###STOP###

# D AIRLINE EXAMPLES

## D.1 DATA EXAMPLES

Here are some examples from `users.json` and `flights.json` respectively. All data is generated by code, and the code is mostly generated by gpt-4, and the gpt-4 prompt is generated by authors.

```json
{
    "name": { "first_name": "Mia", "last_name": "Li" },
    "address": {
        "address1": "975 Sunset Drive",
        "address2": "Suite 217",
        "city": "Austin",
        "country": "USA",
        "province": "TX",
        "zip": "78750"
    },
    "email": "mia.li3818@example.com",
    "dob": "1990-04-05",
    "payment_methods": {
        "credit_card_4421486": {
            "source": "credit_card",
            "brand": "visa",
            "last_four": "7447",
            "id": "credit_card_4421486"
        },
        "certificate_4856383": {
            "source": "certificate",
            "amount": 100,
            "id": "certificate_4856383"
        },
        "certificate_7504069": {
            "source": "certificate",
            "amount": 250,
            "id": "certificate_7504069"
        },
        "credit_card_1955700": {
            "source": "credit_card",
            "brand": "visa",
            "last_four": "1907",
            "id": "credit_card_1955700"
        }
    },
    "saved_passengers": [{ "first_name": "Amelia", "last_name": "Ahmed", "dob": "1957-03-21" }
    "membership": "gold",
    "reservations": ["NO6JO3", "AIXC49", "HKEG34"]
}
```

Listing 4: An example entry from the users database in $\tau$-airline.

```json
{
    "flight_number": "HAT001",
    "origin": "PHL",
    "destination": "LGA",
    "scheduled_departure_time_est": "06:00:00",
    "scheduled_arrival_time_est": "07:00:00",
    "dates": {
        ...,
        "2024-05-12": {
            "status": "landed",
            "actual_departure_time_est": "2024-05-12T05:44:00",
            "actual_arrival_time_est": "2024-05-12T06:39:00"
        },
        "2024-05-13": {
            "status": "landed",
            "actual_departure_time_est": "2024-05-13T05:30:00",
            "actual_arrival_time_est": "2024-05-13T06:32:00"
        },
        "2024-05-14": { "status": "cancelled" },
        "2024-05-15": {
            "status": "landed",
            "actual_departure_time_est": "2024-05-15T06:04:00",
            "actual_arrival_time_est": "2024-05-15T07:30:00"
        },
        "2024-05-16": {
            "status": "available",
            "available_seats": { "basic_economy": 16, "economy": 10,
            ↪   "business": 13 },
            "prices": { "basic_economy": 87, "economy": 122, "business":
            ↪   471 }
        },
        "2024-05-17": {
            "status": "available",
            "available_seats": { "basic_economy": 16, "economy": 13,
            ↪   "business": 9 },
            "prices": { "basic_economy": 76, "economy": 189, "business":
            ↪   498 }
        },
        "2024-05-18": {
            "status": "available",
            "available_seats": { "basic_economy": 2, "economy": 17,
            ↪   "business": 20 },
            "prices": { "basic_economy": 91, "economy": 186, "business":
            ↪   321 }
        },
        ...
    }
}
```

```json
{
    "reservation_id": "4WQ150",
    "user_id": "chen_jackson_3290",
    "origin": "DFW",
    "destination": "LAX",
    "flight_type": "round_trip",
    "cabin": "business",
    "flights": [
        {
            "origin": "DFW",
            "destination": "LAX",
            "flight_number": "HAT170",
            "date": "2024-05-22",
            "price": 883
        },
        {
            "origin": "LAX",
            "destination": "DFW",
            "flight_number": "HAT022",
            "date": "2024-05-26",
            "price": 779
        }
    ],
    "passengers": [
        { "first_name": "Chen", "last_name": "Jackson", "dob": "1956-07-07" },
        { "first_name": "Raj", "last_name": "Smith", "dob": "1967-04-01" },
        { "first_name": "Fatima", "last_name": "Martin", "dob": "1970-01-20" }
    ],
    "payment_history": [{ "payment_id": "gift_card_3576581", "amount": 4986 }],
    "created_at": "2024-05-02T03:10:19",
    "total_baggages": 5,
    "nonfree_baggages": 0,
    "insurance": "no"
}
```

Listing 5: An example entry from the reservations database in $\tau$-airline.

## D.2   TASK 0: SUCCESSFUL TRAJECTORY

Here is a successful gpt-4o agent trajectory, where the agent handles dynamic user intents and complex flight information across the conversation.

```
{
    "instruction": "You are mia_li_3668. You want to fly from New York to
    Seattle on May 20 (one way). You do not want to fly before 11am est.
    You want to fly in economy. You prefer direct flights but one
    stopover also fine. If there are multiple options, you prefer the one
    with the lowest price. You have 3 baggages. You do not want
    insurance. You want to use your two certificates to pay. If only one
    certificate can be used, you prefer using the larger one, and pay the
    rest with your 7447 card. You are reactive to the agent and will not
    say anything that is not asked. Your birthday is in your user profile
    so you do not prefer to provide it.",
    "actions": [
        {
            "name": "book_reservation",
            "arguments": {
                "user_id": "mia_li_3668",
                "origin": "JFK",
                "destination": "SEA",
                "flight_type": "one_way",
                "cabin": "economy",
                "flights": [
                    {
                        "flight_number": "HAT136",
                        "date": "2024-05-20",
                    },
                    {
                        "flight_number": "HAT039",
                        "date": "2024-05-20",
                    },
                ],
                "passengers": [{"first_name": "Mia", "last_name": "Li",
                "dob": "1990-04-05"}],
                "payment_methods": [
                    {"payment_id": "certificate_7504069", "amount": 250},
                    {"payment_id": "credit_card_4421486", "amount": 5},
                ],
                "total_baggages": 3,
                "nonfree_baggages": 0,
                "insurance": "no",
            },
        },
    ],
},
```

**user:** Hello! I am planning a trip and need to book a flight from New York to Seattle. Could you please assist me with this?

**assistant:** Sure, I'd be happy to help you with that. First, I need to get your user ID. Could you please provide it?

**user:** Certainly, my user ID is mia_li_3668.

**assistant:** Great, thank you! Next, could you please provide the following details for your trip:

1. Trip type: One way or round trip?
2. Departure date (and return date if it's a round trip)?

3. Cabin class: Basic economy, economy, or business?
4. Number of passengers and their details (first name, last name, date of birth)?
5. Do you want to purchase travel insurance for this trip (30 dollars per passenger)?

Once I have this information, I can proceed with searching for available flights.

**user**: 1. It's a one-way trip.
2. I want to depart on May 20.
3. I'll be flying in economy class.
4. It's just me, the details should be in my profile.
5. No, I do not want to purchase travel insurance.

**assistant**: Function(arguments='{"user_id":"mia_li_3668"}', name='get_user_details')

**tool**: {"name": {"first_name": "Mia", "last_name": "Li"}, "address": {"address1": "975 Sunset Drive", "address2": "Suite 217", "city": "Austin", "country": "USA", "province": "TX", "zip": "78750"}, "email": "mia.li3818@example.com", "dob": "1990-04-05", "payment_methods": {"credit_card_4421486": {"source": "credit_card", "brand": "visa", "last_four": "7447", "id": "credit_card_4421486"}, "certificate_4856383": {"source": "certificate", "amount": 100, "id": "certificate_4856383"}, "certificate_7504069": {"source": "certificate", "amount": 250, "id": "certificate_7504069"}, "credit_card_1955700": {"source": "credit_card", "brand": "visa", "last_four": "1907", "id": "credit_card_1955700"}}, "saved_passengers": [{"first_name": "Amelia", "last_name": "Ahmed", "dob": "1957-03-21"}], "membership": "gold", "reservations": ["NO6JO3", "AIXC49", "HKEG34"]}

**assistant**: Function(arguments='{"origin":"JFK","destination":"SEA","date":"2024-05-20"}', name='search_direct_flight')

**tool**: [{"flight_number": "HAT069", "origin": "JFK", "destination": "SEA", "scheduled_departure_time_est": "06:00:00", "scheduled_arrival_time_est": "12:00:00", "status": "available", "available_seats": {"basic_economy": 17, "economy": 12, "business": 3}, "prices": {"basic_economy": 51, "economy": 121, "business": 239}}, {"flight_number": "HAT083", "origin": "JFK", "destination": "SEA", "scheduled_departure_time_est": "01:00:00", "scheduled_arrival_time_est": "07:00:00", "status": "available", "available_seats": {"basic_economy": 16, "economy": 7, "business": 3}, "prices": {"basic_economy": 87, "economy": 100, "business": 276}}]

**assistant**: Here are the available direct flights from New York (JFK) to Seattle (SEA) on May 20, 2024:

1. **Flight HAT069**
   - Departure: 06:00 AM EST
   - Arrival: 12:00 PM EST
   - Available Seats: 12 in Economy
   - Price: $121

2. **Flight HAT083**
   - Departure: 01:00 AM EST
   - Arrival: 07:00 AM EST
   - Available Seats: 7 in Economy

        – Price: $100

Please confirm which flight you would like to book.

**user**: Both of the flights are scheduled too early for me. If possible, I would like a flight that does not depart before 11am EST.

**assistant**: Function(arguments='{"origin":"JFK","destination":"SEA","date":"⌐ 2024-05-20"}',
name='search_onestop_flight')

**tool**: [[{"flight_number": "HAT057", "origin": "JFK", "destination": "ATL", "scheduled_departure_time_est": "07:00:00", "scheduled_arrival_time_est": "09:30:00", "status": "available", "available_seats": {"basic_economy": 7, "economy": 3, "business": 9}, "prices": {"basic_economy": 76, "economy": 141, "business": 477}, "date": "2024-05-20"}, {"flight_number": "HAT039", "origin": "ATL", "destination": "SEA", "scheduled_departure_time_est": "22:00:00", "scheduled_arrival_time_est": "03:00:00+1", "status": "available", "available_seats": {"basic_economy": 18, "economy": 10, "business": 2}, "prices": {"basic_economy": 52, "economy": 103, "business": 283}, "date": "2024-05-20"}], [{"flight_number": "HAT136", "origin": "JFK", "destination": "ATL", "scheduled_departure_time_est": "19:00:00", "scheduled_arrival_time_est": "21:30:00", "status": "available", "available_seats": {"basic_economy": 15, "economy": 14, "business": 2}, "prices": {"basic_economy": 52, "economy": 152, "business": 277}, "date": "2024-05-20"}, {"flight_number": "HAT039", "origin": "ATL", "destination": "SEA", "scheduled_departure_time_est": "22:00:00", "scheduled_arrival_time_est": "03:00:00+1", "status": "available", "available_seats": {"basic_economy": 18, "economy": 10, "business": 2}, "prices": {"basic_economy": 52, "economy": 103, "business": 283}, "date": "2024-05-20"}], [{"flight_number": "HAT218", "origin": "JFK", "destination": "ATL", "scheduled_departure_time_est": "18:00:00", "scheduled_arrival_time_est": "20:30:00", "status": "available", "available_seats": {"basic_economy": 12, "economy": 1, "business": 4}, "prices": {"basic_economy": 76, "economy": 158, "business": 342}, "date": "2024-05-20"}, {"flight_number": "HAT039", "origin": "ATL", "destination": "SEA", "scheduled_departure_time_est": "22:00:00", "scheduled_arrival_time_est": "03:00:00+1", "status": "available", "available_seats": {"basic_economy": 18, "economy": 10, "business": 2}, "prices": {"basic_economy": 52, "economy": 103, "business": 283}, "date": "2024-05-20"}], [{"flight_number": "HAT268", "origin": "JFK", "destination": "ATL", "scheduled_departure_time_est": "07:00:00", "scheduled_arrival_time_est": "09:30:00", "status": "available", "available_seats": {"basic_economy": 17, "economy": 19, "business": 12}, "prices": {"basic_economy": 91, "economy": 101, "business": 453}, "date": "2024-05-20"}, {"flight_number": "HAT039", "origin": "ATL", "destination": "SEA", "scheduled_departure_time_est": "22:00:00", "scheduled_arrival_time_est": "03:00:00+1", "status": "available", "available_seats": {"basic_economy": 18, "economy": 10, "business": 2}, "prices": {"basic_economy": 52, "economy": 103, "business": 283}, "date": "2024-05-20"}]]

**assistant**: Here are some one-stop flight options from New York (JFK) to Seattle (SEA) on May 20, 2024, with departure times after 11:00 AM EST:

1. **Flight Option 1**
   – **First Leg:**
     – Flight Number: HAT136
     – Departure: 07:00 PM EST from JFK
     – Arrival: 09:30 PM EST at ATL
     – Available Seats: 14 in Economy

         - Price: $152
     - **Second Leg:**
       - Flight Number: HAT039
       - Departure: 10:00 PM EST from ATL
       - Arrival: 03:00 AM EST (+1 day) at SEA
       - Available Seats: 10 in Economy
       - Price: $103

2. **Flight Option 2**
     - **First Leg:**
       - Flight Number: HAT218
       - Departure: 06:00 PM EST from JFK
       - Arrival: 08:30 PM EST at ATL
       - Available Seats: 1 in Economy
       - Price: $158
     - **Second Leg:**
       - Flight Number: HAT039
       - Departure: 10:00 PM EST from ATL
       - Arrival: 03:00 AM EST (+1 day) at SEA
       - Available Seats: 10 in Economy
       - Price: $103

Please confirm which flight option you would like to book.

**user:** I'll go with the first option: Flight HAT136 departing from JFK at 07:00 PM EST and Flight HAT039 departing from ATL at 10:00 PM EST.

**assistant:** Here are the details for your booking:

- **Trip Type:** One-way
- **Origin:** New York (JFK)
- **Destination:** Seattle (SEA)
- **Cabin Class:** Economy
- **Flights:**
  - **First Leg:**
    - Flight Number: HAT136
    - Departure: 07:00 PM EST from JFK
    - Arrival: 09:30 PM EST at ATL
  - **Second Leg:**
    - Flight Number: HAT039
    - Departure: 10:00 PM EST from ATL
    - Arrival: 03:00 AM EST (+1 day) at SEA
- **Passenger:**
  - Mia Li (DOB: 1990-04-05)
- **Total Price:** $152 + $103 = $255
- **Checked Bags:** 3 free checked bags (Gold member in economy class)
- **Travel Insurance:** No

**Payment Methods:**
- You have the following payment methods available:
  - Credit Card ending in 7447
  - Credit Card ending in 1907
  - Certificate with $100
  - Certificate with $250

Please confirm if you would like to proceed with this booking and specify the payment method(s) you would like to use.

**user:** Yes, the booking information is correct. For the payment, I would like to use my two certificates first. If only one certificate can be used, please use the one with $250, and charge the rest to the credit card ending in 7447.

**assistant:** For safety reasons, only one travel certificate can be used per reservation. Therefore, I will use the $250 certificate and charge the remaining amount to the credit card ending in 7447.

Here are the payment details:
- **Certificate:** $250
- **Credit Card (ending in 7447):** $5

Please confirm if you would like to proceed with this payment arrangement.

**user:** Yes, I confirm. Please proceed with the payment arrangement.

**assistant:** Function(arguments='{"user_id":"mia_li_3668","origin":"JFK","des⌡
tination":"SEA","flight_type":"one_way","cabin":"economy","flights":[{"f⌡
light_number":"HAT136","date":"2024-05-20"},{"flight_number":"HAT039","d⌡
ate":"2024-05-20"}],"passengers":[{"first_name":"Mia","last_name":"Li","⌡
dob":"1990-04-05"}],"payment_methods":[{"payment_id":"certificate_750406⌡
9","amount":250},{"payment_id":"credit_card_4421486","amount":5}],"total⌡
_baggages":3,"nonfree_baggages":0,"insurance":"no"}',
name='book_reservation')

**tool:** {"reservation_id": "HATHAT", "user_id": "mia_li_3668", "origin": "JFK", "destination": "SEA", "flight_type": "one_way", "cabin": "economy", "flights": [{"flight_number": "HAT136", "date": "2024-05-20", "price": 152, "origin": "JFK", "destination": "ATL"}, {"flight_number": "HAT039", "date": "2024-05-20", "price": 103, "origin": "ATL", "destination": "SEA"}], "passengers": [{"first_name": "Mia", "last_name": "Li", "dob": "1990-04-05"}], "payment_history": [{"payment_id": "certificate_7504069", "amount": 250}, {"payment_id": "credit_card_4421486", "amount": 5}], "created_at": "2024-05-15T15:00:00", "total_baggages": 3, "nonfree_baggages": 0, "insurance": "no"}

**assistant:** Your flight has been successfully booked! Here are the details of your reservation:

- **Reservation ID:** HATHAT
- **Trip Type:** One-way
- **Origin:** New York (JFK)
- **Destination:** Seattle (SEA)
- **Cabin Class:** Economy
- **Flights:**
  - **First Leg:**
    - Flight Number: HAT136
    - Departure: 07:00 PM EST from JFK
    - Arrival: 09:30 PM EST at ATL
  - **Second Leg:**
    - Flight Number: HAT039
    - Departure: 10:00 PM EST from ATL
    - Arrival: 03:00 AM EST (+1 day) at SEA
- **Passenger:**
  - Mia Li (DOB: 1990-04-05)
- **Checked Bags:** 3 free checked bags
- **Travel Insurance:** No

**Payment Details:**
- **Certificate:** $250
- **Credit Card (ending in 7447):** $5

If you have any further questions or need additional assistance, feel free to ask. Have a great trip!

**user:** ###STOP###

