# OpenReview forum: "{$\tau$}-bench: A Benchmark for \underline{T}ool-\underline{A}gent-\underline{U}ser Interaction in Real-World Domains"
_ICLR.cc/2025/Conference — ICLR 2025 Poster_

### Official Review · Reviewer_D4Hj · 2024-11-01

**Soundness:** 4
**Presentation:** 4
**Contribution:** 4
**Rating:** 8
**Confidence:** 4

**Summary:**

This paper proposes Tau-Bench, a benchmark for measuring the performance of models on user/agent interactions where the model has the ability to use tools. The goal of each interaction is to learn information from the user, and achieve some end state the user wants the agent to accomplish, such as re-booking a flight or initiating a return through an online store. They also introduce a pass^k (similar to pass@k) metric, measuring how often an agent succeeds at a given task when the task is re-run with slight variations in syntax/semantics/etc, simulating multiple users attempting to accomplish the same goal, to measure how reliable an agent is.

**Strengths:**

This paper proposes a novel benchmark targeting a growing area of interest (user/agent interactions to achieve a goal), and provides a strong foundation for future work in this area. The benchmark is well-defined and easy to understand, the proposed pass^k metric seems like a great measure of reliability across a diverse set of users and scenarios, and the tasks in the benchmark are focused on popular, real-world, tasks that are difficult for current models.

**Weaknesses:**

I think this paper is very strong overall. There are a few additional results I'd be interested to see, but I don't think any are necessarily required:
* how consistent is pass^k? E.g. if you run pass^16 10 times, what is the variance?
* what is the effect of changing the policy document? does model performance change much with more, fewer, or different constraints?
* do in context examples help improve agent consistency?

**Questions:**

My one suggestion would be to explore (in the final revision, or follow up work) how well open models can do when finetuned specifically for these sorts of tasks. The formulation of agent trajectories in this work would lend itself very well to both supervised finetuning on example gold interactions, and preference tuning on positive and negative examples. Because the answers are verifiable, reinforcement learning would also be a good candidate. I'm curious how well specialized models can perform these tasks, especially compared to strong closed models such as gpt4o.

---

> ### Author Response · Authors · 2024-11-21
>
> Thanks for finding our work “novel” and “very strong overall”!
>
> # consistency of pass^k
>
> It depends on $n$, the number of trials you run a task. If you run a task for $n$ trials to calculate pass^k, you are averaging $n \choose k$ samples of pass^k (each “sample” is a choice of k of the n trials), so the variance will be divided by $n \choose k$. For example, if you run 20 trials to calculate pass^16, the variance will be around 1/5000 of the variance of using 16 trials (i.e. the naive estimator). If you use 30 trials, the variance will be around less than $10^{-8}$ times the naive estimator variance. In summary, even using $n$ slightly larger than $k$ will significantly reduce the variance of pass^k estimation.
>
> It also depends on your number of tasks. If you have 100 tasks in the benchmark, the variance will be 1/100 of the variance if you only have a single task in the benchmark.
>
> We will revise the draft to discuss these important statistical properties of pass^k.
>
> # changing policy document
>
> Unfortunately, we cannot easily change the policy document, as the APIs, database schema, and tasks are now manually co-designed with the policy. If the policy changes, the ground truth actions for a task might also change, making evaluation intractable.
>
> # few-shot agents
>
> Good question! Following your suggestion, we did additional 5-shot prompting experiments over 4 trails, where we used gpt-4o tool-calling with domain-specific few-shots (retail few-shots for retail tasks, airline few-shots for airline tasks) and cross-domain few-shots (airline few-shots for retail tasks, retail few-shots for airline tasks). For each domain, we collect ~20 successful trajectories, and during testing randomly select 5 trajectories that are not from the same task. Results are as follows:
>
> * Retail acc (baseline = 60.4%)
>    * Domain-specific few-shots = 65.2%
>    * Cross-domain few-shots = 61.5%
>
> * Airline (baseline = 42.0%)
>    * Domain-specific few-shots = 45.5%
>    * Cross-domain few-shots = 42.5%
>
> Based on the results, adding few-shot examples improves performance across the retail and airline domains. In addition, domain-specific few-shots are more effective than cross-domain few-shots. But few-shot is significantly more costly than zero-shot due to the length of 5 trajectories, so there is a cost-performance tradeoff. We will include these results in revision.
>
> # fine-tuning agents
>
> We agree this is an important and exciting future research! One caveat is that the size of tau-bench now is still small, due to the limits of manual API, database, policy, and task design. Future research into automatic construction of scalable training tasks might help unlock agent fine-tuning or RL in this case.

---

### Official Review · Reviewer_p3GC · 2024-11-03

**Soundness:** 2
**Presentation:** 4
**Contribution:** 2
**Rating:** 6
**Confidence:** 4

**Summary:**

The paper proposes a new LLM-agent benchmark for two domains (retail and airline), called tau-bench. The gap that the proposed tau-bench aims to fill is LLM-agent benchmarks that focus on User and LLM agent interactions. I.e. measuring how well a LLM agent performs when interacting with a "human" user. To this end they propose to use a LLM based user simulator. Furthermore, the paper describes how the benchmark is constructed and based on databases and APIs. Finally, the authors also propose a new metric for the benchmark, pass HAT k (pass ^ k). This metric measures what the chance is that all passes are successful given k i.i.d. attempts.

The paper has a detailed evaluation of agents performance on the benchmark, detailed analysis of errors of these agents and some cost analysis of both the agent (LLM-based) and user-simulator (LLM-based). The cost of a single evaluation cost $200. There is also some analysis of user simulator performance.

**Strengths:**

The paper is very clear, well written and well designed. The paper shows that much effort has been put into this work.

Important contributions:
1. The main contribution is the benchmark itself. Specifically, the databases, function calls and interaction templates.

2. The evaluation of baseline agents and ablation studies are interesting to see.

3. The proposal of the metric pass ^ k (pass hat k, vs. the existing metrics pass @ k) is quite interesting.

4. Good comparison against previous work, including task oriented dialogue systems (well done).

**Weaknesses:**

The main weaknesses of the paper is the analysis of the **user simulator**, evaluation of it, and especially using less *prohibitively expensive* models such as open source model of the size 7B-13B.

1. The paper evaluated the user simulator based on mistakes of agents (and identified 4% of errors). However, it would be interesting to establish guarantees on the user simulator.
2. Additionally, it would be important to use such evaluation benchmarks of the user simulator to show how smaller models perform as user simulators to make such a benchmark more usable and available to the community.
3. Therefore, at this stage it is not clear at all how smaller models would perform as user simulators, whether it is possible or not.
4. Also, having a benchmark that only works with the highest end APIs is very prohibitive for the research community. (Great projects that could help alleviate this problem for you are: AutoGuide (a RAG mechanism), Act-Re (a finetuning mechanism), StateAct (a prompting mechanism), Optimus-1 (a finetuning & exploration mechanism)).

---
An additional weakness is the presentation of the *unbiased estimator* of the pass ^ k (pass hat k) metric is unclear. There is no derivation or explanation where this formula comes from and why it should be an unbiased estimator.

**Questions:**

1. How do you evaluate the user simulator?
2. How could you evaluate the user simulator?
3. What would be the performance of a smaller LLM (7B / 13B) as user simulator?
4. How do you derive the unbiased estimator?

---

> ### Author Response · Authors · 2024-11-21
>
> Thank you for finding our work “well-written”, “well-designed”, and “interesting”!
>
> # guarantees and evaluation of the user simulator
>
> See General Response.
>
> # small models and cost concerts for user simulator
>
> To address your concern that user simulation might be expensive and your question of whether small models can be good user simulation, we perform additional experiments over 4 trails using gpt-4o-mini as the user simulator, and the comparison to gpt-4o as user simulator is as follows:
>
> * gpt-4o: $0.399/task
>    * Retail: agent pass^1 = 60.4%, user simulation error = 2%
>    * Airline: agent pass^1 = 42.0%, user simulation error = 2%
>
> * gpt-4o-mini: $0.024/task
>    * Retail: agent pass^1 = 52.3%, user simulation error = 11%
>    * Airline: agent pass^1 = 32.0%, user simulation error = 13.2%
>
> According to our analysis, using a smaller model such as gpt-4o-mini might cost 10x less, but the quality of the user simulator will deteriorate significantly. One can expect Llama 7B/13B be even worse than gpt-4o-mini for user simulation. Thus, we’d still recommend using gpt-4o as user simulator for now. We will revise the draft based on these findings.
>
> # RAG/fine-tuning/prompting
>
> We agree developing new methods to solve tau-bench is important future work, and there is hope these methods you mentioned can utilize open models to solve tau-bench with improved performances and lowered costs.
>
> # Unbiased estimator
>
> When you run a task for $n$ trials to estimate pass^k, there are $n \choose k$ possible subsets of $k$ trials, of which there are only $c \choose k$ possible subsets where pass^k=1 (where $c$ is the number of successful trials), and the only subsets have pass^k=0. Therefore averaging over all these subsets, the expected value of pass^k is ${c \choose k} / {n \choose k}$. We will better explain this in revision.

---

> > ### Comment · Reviewer_p3GC · 2024-11-26
> >
> > Dear authors,
> >
> > Thank you for the response and running additional experiments using smaller models and confirming the results and costs. Also, thank you for the unbiased estimator explanation.
> >
> > (Apologies for our delayed response, due to high work-load at the moment.)
> >
> > Clarification question, when you refer to:
> >
> > > gpt-4o-mini: $0.024/task
> >
> > What do you mean by task? And what is the total cost of running the Retail domain / Airline domain?
> >
> > ---
> > Overall, we raise the score, as we believe this work counts as a valid contribution and is novel. We still believe it will not be applicable to many researchers due to the cost of running such user simulators - however, we hope future work will focus also on OS models that can produce strong user simulators.

---

> > > ### Author Response · Authors · 2024-11-27
> > >
> > > Thank you for your response and acknowledgement in this busy time! By task we mean each task instance in the domain, not each domain. We agree, future research to improve user simulation with os model will be very impactful and make tau bench more accessible.

---

### Official Review · Reviewer_z9mi · 2024-11-04

**Soundness:** 3
**Presentation:** 3
**Contribution:** 3
**Rating:** 6
**Confidence:** 4

**Summary:**

the paper build a novel agent evaluation benchmark considering the complexity of domain policy, user-agent interactions with  realistic databases and APIs. Based on that, the paper propose new evalaution metric pass^k to determine the robust and consistency of agent, and present insightful analysis about existing LLMs.

**Strengths:**

1. the target problem is valuable and important, such as the  complex policies and rules specific to a task or domain that the agent should follow, and consistency and reliability at scale.
2. the formulation is reasonable and proposed collection pipeline is easy to understand and expand
3. the experiment and analysis are comprehensive and insightful.

**Weaknesses:**

1. the contribution is a little overclaimed and the paper does not given enough credits to task-oriented dialogue system (ToD). There are several examples: 1) evaluation scheme in line 78, the dialogue state tracking in ToD also compare the current dialogue state with ground truth expected state; 2) there are many lm-simulated users in ToD (see below) and rule-based user simulator also bring insightful studies; 3) some important works are missing, such schema-guided dialogue dataset which also explore various schema in different domain and build corresponding APIs and collect data via human-to-human ways. Generally, the major contribution of this paper lies in combination of lm-simulated user, real APIs and introduction of domain-specific policy while others such as multi-turn interactions are very similar with ToD.

2. the related work is not fully discussed. 1) line 108-111, there are many works focus on multi-step user interaction, such as API-Bank [1], and ToolUltra [2]; 2) there are many LM-simulated user studies [3,4,5].


[1] API-Bank: A Comprehensive Benchmark for Tool-Augmented LLMs
[2] Planning, Creation, Usage: Benchmarking LLMs for Comprehensive Tool Utilization in Real-World Complex Scenarios
[3] Beyond Static Evaluation: A Dynamic Approach to Assessing AI Assistants' API Invocation Capabilities
[4] Reliable LLM-based User Simulator for Task-Oriented Dialogue Systems
[5] PlatoLM: Teaching LLMs in Multi-Round Dialogue via a User Simulator

**Questions:**

1. line 349, why self-reflection is not suitable? despite the agent only have one change to serve the user, it still can reflect its own reasoning and API call before it generate the final responses.

---

> ### Author Response · Authors · 2024-11-20
>
> Thanks for finding our work important, comprehensive, and insightful!
>
> # credits to task-oriented dialogue system (ToD)
>
> Thank you for pointing out these important prior works relevant to us in various aspects! You are totally right that “the major contribution of this paper lies in combination of lm-simulated user, real APIs and introduction of domain-specific policy” (see lines 49-50 of our work) — our benchmark inherits ideas from prior benchmarks of both autonomous (language) agents and ToD, and can be seen as an intersection of both. We will revise our draft to cite and discuss these works, and stress that
>
> - our evaluation scheme (line 78) resembles state tracking in ToD (with references to prior ToD works)
> - our main contribution is NOT the proposal of LLM-based user simulation on its own (with references to your citations), but rather its combination with real-world policy, APIs, and tasks for agent benchmarking.
>
> # why self-reflection is not suitable
>
> To clarify, we meant self-reflection **after** receiving the observation from users or tools, which is the setup of Reflexion (Shinn et al., 2023) cited in line 349. This is intuitive — while game or QA agents studied in Reflexion (Shinn et al., 2023) can reset the game or web browser to “try again” with self-reflection, a customer service agent facing customers cannot “reset” the human if something bad or offensive is uttered to the user. We will better clarify this point in the draft revision.

---

### Official Review · Reviewer_EMki · 2024-11-04

**Soundness:** 3
**Presentation:** 3
**Contribution:** 3
**Rating:** 6
**Confidence:** 3

**Summary:**

This article introduces a novel benchmark called τ-bench, designed to evaluate the ability of language agents to interact and plan with human users and programmatic APIs in real-world scenarios. The main contributions of the article can be summarized as follows:
- **Introduction of the τ-bench Benchmark**: τ-bench simulates dynamic dialogues between users and language agents equipped with domain-specific API tools and strategy guidelines. It is built on a modular framework, featuring realistic databases and APIs, domain-specific strategy documents, and detailed instructions for various user scenarios, along with corresponding real annotations.
- **New Evaluation Metric, passˆk**: In addition to measuring the success rate of agents in task completion (passˆ1), the article proposes a new metric, passˆk, to assess the consistency and robustness of agent behavior across multiple independent trials. This metric is particularly important for real-world agent tasks requiring reliability and consistency, such as customer service.
- **Comprehensive Experiments on Existing Language Models**: The authors conduct extensive tests using τ-bench on a variety of state-of-the-art proprietary and open-source language models, including GPT, Claude, Gemini, Mistral, and Llama. The experimental results indicate that even the most advanced functional calling agents (e.g., gpt-4o) succeed in less than 50% of tasks and exhibit poor consistency (with passˆ8 < 25% in the retail domain).

**Strengths:**

1. The τ-bench benchmark effectively addresses the limitations of existing evaluations by providing a more comprehensive framework for assessing the capabilities of language agents in real-world environments. The introduction of the passˆk metric presents a novel approach to measuring the consistency and robustness of agent behavior, while the pass in k demonstrates the potential for using agents in data synthesis.
2. The dataset construction closely aligns with practical application scenarios. The testing methods and data definitions exhibit scalability, enhancing the utility of the benchmark for future research.
3. The experiments are comprehensive. The article is well-written and easy to follow.
4. The qualitative experiments are highly detailed, pointing out promising directions for future research.

**Weaknesses:**

1. **Limitations of user simulation**: The user behaviors are simulated by LLMs in this study. Although the authors explore methods to improve user simulation, these efforts cannot entirely mitigate the potential issues arising from simulated users. It may result in simulated behaviors that diverge from those of real users, thereby affecting the assessment of the agents.
2. **Limitations of Evaluation Metrics**: While the passˆk metric assesses the consistency and robustness of agent behavior, it remains a task success rate-based measure. In some cases, an agent may provide valuable information or service even if it does not fully accomplish the intended task. It may make more sense to use more granular evaluation metrics.

**Questions:**

This evaluation framework is likely to benefit from more fine-grained task definition and evaluation, such as information gathering, policy understanding, decision-making, etc., assessing the agent's performance on each subtask. This will provide more comprehensive understanding of the agent's capabilities, offering specific guidance for future research. Additionally, the paper could incorporate more dimensions of human evaluation, such as the agent's understanding of user intent, adherence to policy, and user satisfaction with the agent.

Other questions can refer to the weaknesses.

---

> ### Author Response · Authors · 2024-11-21
>
> Thanks for finding our work practical, comprehensive, pointing out promising directions for future research!
>
> # Limitations of user simulation
>
> Check General Response.
>
> # Limitations of evaluation metrics
>
> We totally agree that more fine-grained metrics will reveal more insights into agent behaviors, but there are some challenges:
>
> - Metrics based on heuristics (e.g. distance to goal, tool call F1 against GT actions) might not correlate with any meaningful aspects (e.g. policy following, user following), and optimizing for these might be less meaningful than success rate or even harmful.
>
> - Metrics based on trajectory examination might tell more fine-grained aspects, e.g policy following, user following, etc. However, using human examination is expensive, and using LLM eval is not guaranteed to be reliable, requiring future research.
>
> We will add these discussions to revision and highlight this as a limitation and important future direction of our work.

---

### Author Response · Authors · 2024-11-21
**General Response**

We appreciate the positive feedback from reviewers!

# Our contributions and enabled future directions


Our work has several important contributions:
- The benchmark itself, and its detailed design principles and implementations
- The idea to **combine** (we acknowledge prior work for each aspect individually, but only combining all three can lead to tasks with realistic tool-agent-user interactions) real-world APIs, domain-specific policy following, and user simulation, which we believe are three key gaps of today’s agent benchmarks from real-world applications.
- The introduction of pass^k metric for reliability and robustness, which can be used in other tasks
- Comprehensive experiments across various models and various analyses

Our current work already has vast content, details, and workloads (acknowledged by all reviewers), so we could not pursue many interesting (and natural) followup directions from tau-bench, which we have to leave as future research:

- Better user simulation methods, and better ways to evaluate user simulation in tool-agent-user interactions; see the next section for more details. (EMki, p3GC)
- Better agent methods (e.g., RAG, fine-tuning, few-shot prompting) to handle tool-agent-user interactions (p3GC, D4Hj)
- Better metrics for fine-grained evaluation of tool-agent-user interactions (EMki)
- Better ways to automatically and scalably construct benchmarks for tool-agent-user interactions, beyond manual design of APIs/policy/tasks

**Note that none of these directions are possible to pursue before the introduction of tau-bench.** So as stated by reviewer D4Hj, our work “provides a strong foundation for future work in this area”, and we are excited to see followup research in these various exciting directions.


# User simulation and its evaluation

Unfortunately, LLM-based user simulators cannot be guaranteed to always work, **just like no other LLM-based systems (or even real humans) can be guaranteed to never hallucinate**. Instead of formal guarantees, we believe it is much more tractable (and indeed extremely important) to establish empirical evaluations of user simulation. We have shown two approaches in our work (see Section 5.3):

1. (Main) Human examination of trajectories, where we find our user simulation currently only has a low error rate of 4%.

2. Controlled experiments with the same agent paired with different user simulators. In Table 4, we find that using slightly more complicated prompting methods (e.g., reflection, verify) slightly improves the agent-user collaborative performances, but not a lot (partly because user error rate is already low in current tau-bench). Our additional experiments using gpt-4o-mini as user simulator also shows that a weaker user simulation model will lead to lowered agent performance and increased user errors, so agent performance could be a noisy proxy to user simulation performance.

* gpt-4o: $0.399/task
   * Retail: agent pass^1 = 60.4%, user simulation error = 2%
   * Airline: agent pass^1 = 42.0%, user simulation error = 2%

* gpt-4o-mini: $0.024/task
   * Retail: agent pass^1 = 52.3%, user simulation error = 11%
   * Airline: agent pass^1 = 32.0%, user simulation error = 13.2%


**In summary, for today’s tau-bench tasks, which are small-scale and relatively simple, we find that user simulation has a very low error rate, and human examination can be effectively used to evaluate user simulation.** However, if future tasks evolve to be large-scale and significantly more complex, we’d expect today’s models and naive methods for user simulation to have more errors, and human examination to be less tractable for evaluation. In such a future case, we believe it is important to study better ways to construct and evaluate user simulation. We will revise our draft and incorporate such a discussion.

---

### Comment · Reviewer_p3GC · 2024-12-01
**Comment after Reviews**

Overall, this work is very interesting and a good contribution.

The main concern remains the wider usability. However, we also acknowledge that this can be directed to future work.

---

### Meta-Review · Area_Chair_SMQz · 2024-12-23

**Metareview:**

The paper "$\tau$-bench" introduces $\tau$-bench, a benchmark for evaluating LM interactions with users and tools in domains like retail and airlines. It combines APIs, realistic databases, and policy adherence with a novel pass^k metric for reliability. Comprehensive experiments show current models struggle with consistency and rule-following, highlighting future research needs.

Strengths include the innovative benchmark design, the novel metric, and thorough experiments. Weaknesses involve overclaimed novelty, limited related work citations, and user simulation costs. Despite this, the benchmark’s utility for future work makes it recommendable for acceptance.

**Additional Comments On Reviewer Discussion:**

Reviewers raised concerns about user simulation, novelty claims, and evaluation metrics. Authors addressed these by revising related work citations, conducting additional experiments with smaller LMs, and discussing cost-performance trade-offs. While issues like accessibility remain, the strong rebuttals and revisions justify acceptance.

---

### Decision · Program_Chairs · 2025-01-22

Accept (Poster)